J Physiol 600.18 (2022) pp 4069–4087

TOPICAL REVIEW

# Differential contributions of cardiac, coronary and pulmonary artery vagal mechanoreceptors to reflex control of the circulation

Jonathan P. Moore[1] 🆔, Lydia L. Simpson[2] 🆔 and Mark J. Drinkhill[3] 🆔

[1] School of Human and Behavioural Sciences, Bangor University, Bangor, UK
[2] Department of Sport Science, University of Innsbruck, Innsbruck, Austria
[3] Leeds Insititute for Cardiovascular and Metabolic Medicine, University of Leeds, Leeds, UK

Handling Editors: Ian Forsythe & Emma Hart

The peer review history is available in the Supporting Information section of this article (https://doi.org/10.1113/JP282305#support-information-section).

**Abstract** Distinct populations of stretch-sensitive mechanoreceptors attached to myelinated vagal afferents are found in the heart and adjoining coronary and pulmonary circulations. Receptors at atrio-venous junctions appear to be involved in control of intravascular volume.

**Jonathan Moore** (pictured) and **Mark Drinkhill** are senior lecturers and researchers interested in neural control and autonomic regulation of the circulation. Both were awarded their PhDs from the University of Leeds, and both completed postdoctoral training under the guidance of Roger Hainsworth. **Lydia Simpson** completed her PhD at Bangor University. Currently she is a postdoctoral fellow at the University of Innsbruck.

These atrial receptors influence sympathetic control of the heart and kidney, but contribute little to reflex control of systemic vascular resistance. Baroreceptors at the origins of the coronary circulation elicit reflex vasodilatation, like feedback control from systemic arterial baroreceptors, as well as having characteristics that could contribute to regulation of mean pressure. In contrast, feedback from baroreceptors in the pulmonary artery and bifurcation is excitatory and elicits a pressor response. Elevation of pulmonary arterial pressure resets the vasomotor limb of the systemic arterial baroreflex, which could be relevant for control of sympathetic vasoconstrictor outflow during exercise and other states associated with elevated pulmonary arterial pressure. Ventricular receptors, situated mainly in the inferior posterior wall of the left ventricle, and attached to unmyelinated vagal afferents, are relatively inactive under basal conditions. However, a change to the biochemical environment of cardiac tissue surrounding these receptors elicits a depressor response. Some ventricular receptors respond, modestly, to mechanical distortion. Probably, ventricular receptors contribute little to tonic feedback control; however, reflex bradycardia and hypotension in response to chemical activation may decrease the work of the heart during myocardial ischaemia. Overall, greater awareness of heterogeneous reflex effects originating from cardiac, coronary and pulmonary artery mechanoreceptors is required for a better understanding of integrated neural control of circulatory function and arterial blood pressure.

(Received 18 October 2021; accepted after revision 19 July 2022; first published online 28 July 2022)

**Corresponding author** J. Moore: School of Human and Behavioural Sciences, Bangor University, Bangor LS57 2PZ, UK.
Email: j.p.moore@bangor.ac.uk

**Abstract figure legend** A schematic illustration of neural inputs to the cardiovascular control centre. The schema provides examples of typical afferent discharge, and related pressure traces, at several locations (*A*) and depicts integration of vagal afferent signals arising from systemic arterial baroreceptors, pulmonary arterial baroreceptors, atrial volume receptors, coronary arterial receptors and ventricular receptors (*B*). Impulse activity recorded from a pulmonary arterial receptor displays the familiar pattern associated with arterial baroreceptors. Atrial pattern type A displays a volley of impulses that corresponds with atrial systole and a decrease in atrial volume, whereas type B pattern corresponds to atrial diastole and filling of the atrium. Discharge from a coronary mechanoreceptor begins to rise before the aortic pressure and this corresponds the coronary perfusion pulse. Discharge from an unmyelinated vagal afferent increases briskly following injection of veratridine into the aortic root. The table summarizes fibre type, discharge pattern, activating stimuli and reflex effects. *Increase in vagal activity to the heart with no effect on sympathetic activity. #Increase in sympathetic activity to the heart with no effect on vagal activity or inotropy.

Complex sensory nerve endings, functioning as stretch receptors, exist within both the walls of the heart and the walls of the adjoining systemic, coronary and pulmonary arterial systems. The regulation of arterial blood pressure by high pressure baroreceptors located in the ascending aorta and carotid sinuses is understood reasonably well (Kirchheim, 1976; Sagawa, 1983). In contrast, the complexity of reflex control by receptors located in other areas, such as the atrio-venous junctions (Coleridge et al., 1957), cardiac ventricles (Coleridge et al., 1964), coronary arteries (Brown, 1965) and extrapulmonary portion of the pulmonary artery (Bianconi & Green, 1959), is more difficult to comprehend. These mechano-receptors are referred to collectively as 'cardiopulmonary' receptors (Heymans & Neil, 1958). Moreover, despite spatial heterogeneity and differences in stimulation, it is accepted that cardiopulmonary afferent pathways are responsible for an inhibitory effect on peripheral vascular tone and cardiac function, like negative feedback arising from systemic arterial baroreceptors (Fig. 1).

For this review, we concentrate on the reflexes mediated by vagal afferents arising from atrial, ventricular, coronary and pulmonary arterial mechanoreceptors. Circulatory reflexes mediated by afferents running in the sympathetic trunk (Brown, 1967; Brown & Malliani, 1971; Malliani et al., 1986) are beyond the scope of this review and are not discussed further. We consider some general aspects of vagal reflexes, including whether the receptors are attached to myelinated or unmyelinated afferents, the nature of effective stimuli, and whether reflexes are inhibitory or excitatory. Much of the primary mechanistic evidence comes from experiments in animals that employ direct neural recording and/or vascular isolation and perfusion techniques. By comparison, interventions used to study 'cardiopulmonary' mechanoreceptor reflexes in humans, such as lower body suction, tend

to have limited specificity. Nonetheless, it is frequently reported that 'cardiopulmonary' receptor unloading elicits excitatory effects (Park et al., 2018), whereas activation of 'cardiopulmonary' receptors elicit an inhibitory reflex (Katayama et al., 2020). Recently, however, there have been suggestions that human muscle sympathetic vasoconstrictor nerves may be excited by atrial filling (Incognito et al., 2019; Millar et al., 2013) and by elevated pulmonary arterial pressure (Simpson et al., 2020). This divergence from the traditional view suggests a more nuanced view of cardiopulmonary reflexogenic areas may be required. To develop this, we integrate evidence from animal experiments with that from human studies, and we propose an updated model that acknowledges differential neural control by atrial, coronary and pulmonary arterial mechanoreceptors.

## Afferent vagal nerves from the heart and the coronary and pulmonary arteries

**Atrial afferents.** The upper chambers of the heart are supplied with myelinated vagal afferents attached to mechanoreceptors that have been localized primarily to atrio-venous junctions (Coleridge et al., 1957). The activity pattern of these receptors is classified as type A, type B or intermediate, depending upon when the peak discharge appears in relation to the atrial pressure wave (Paintal, 1953, 1963). Type A pattern corresponds to the 'a' wave of atrial systole, type B corresponds to the 'v' wave of atrial filling, and the intermediate type has characteristics of both types A and B. It is generally accepted that a single type myelinated atrial receptor exists, but the discharge pattern varies according to the location of the receptor in the atrial wall (Kappagoda et al., 1976).

The effective stimulus for atrial myelinated vagal afferents is wall distension, which is determined primarily by atrial volume (Kidd et al., 1978). Furthermore, atrial size might influence discharge, as Hicks and colleagues found atrial receptor discharge was greater in large dogs than in small dogs, and discharge in small dogs was greater than for cats (Hicks et al., 1990). Non-myelinated vagal fibres with mechanosensitive endings in the atrial walls have also been identified, and there appear to be many more of these compared with myelinated nerves (Coleridge et al., 1973; Thoren, 1976). However, discharge is sparse, and unmyelinated afferents display a higher

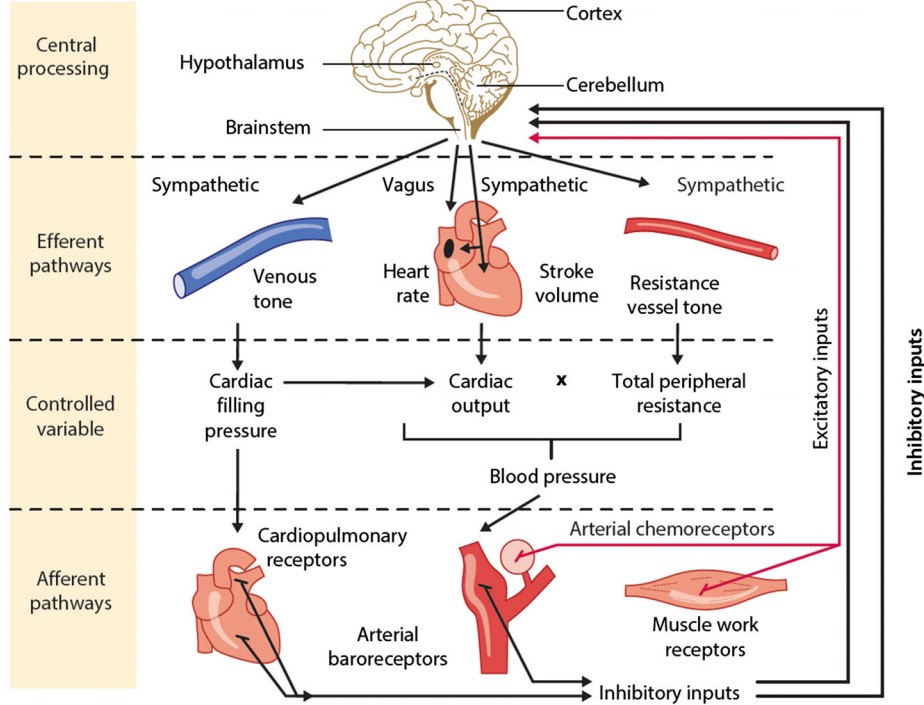

**Figure 1. Overview of neural reflex control of the circulation**
Inhibitory and excitatory refer to the net effect on blood pressure. Inhibitory reflexes are depressor and excitatory reflexes are pressor. Input from cardiopulmonary receptors is described uniformly as inhibitory despite evidence to the contrary. Reproduced from *Levick's Introduction to Cardiovascular to Physiology*, 6th Edition, Herring & Paterson (2018). © 2018 by Taylor & Francis Group, LLC. Reproduced with the permission of the Licensor through PLSclear.

threshold for activation compared with myelinated atrial receptors.

**Ventricular vagal afferents.** Afferent innervation of the cardiac ventricles is dominated by unmyelinated vagal nerve fibres, which originate mostly from receptors located in the epicardium and myocardium of the posterior wall of the left ventricle (Coleridge et al., 1964; Sleight & Widdicombe, 1965). Direct neurophysiological recordings from ventricular afferents in dogs and cats reveal periods of neural silence interspersed by low frequency irregular discharge (Coleridge et al., 1964; Oberg & Thoren, 1972a, 1972b; Thames et al., 1977; Thorén, 1977). Because most unmyelinated ventricular vagal afferents discharge sporadically, without any obvious relation to normal ventricular haemodynamic events, it is not clear how this input signal contributes to normal neural circulatory control.

Most unmyelinated ventricular afferents respond overtly with a rapid burst of activity to chemical irritants, such as veratridine, phenyl diguanide or capsaicin, applied directly to receptor endings or injected into the coronary circulation (Coleridge et al., 1964; Drinkhill et al., 1993; Sleight & Widdicombe, 1965). Furthermore, prostaglandins and bradykinin, which are formed and released by stressed myocardial tissue, strongly stimulate many ventricular afferents (Kaufman et al., 1980; Schultz & Ustinova, 1996; Schultz et al., 1997). However, considerable variability exists in relative sensitivity to various chemical mediators.

Some ventricular afferents develop a rhythmic pattern of discharge during occlusion of the ascending aorta (Coleridge et al., 1964; Drinkhill et al., 1993; Thorén, 1977), blood volume expansion (Oberg & Thoren, 1972a, 1972b), or an increase in ventricular inotropic state (Muers & Sleight, 1972; Sleight & Widdicombe, 1965). Mechanosensitive afferents appear to be activated most by such manoeuvres during ventricular systole (Muers & Sleight, 1972; Thorén, 1977). However, frequency of discharge appears to correlate well with diastolic events (Thorén, 1977). Notably, distension of a fibrillated heart does not alter receptor activation. On the other hand, changes in ventricular contractility appear to alter responses to distension by aortic occlusion, or increases in blood volume (Thorén, 1977). Thus, the effective stimulus to mechanosensitive ventricular receptors is unclear. Possibly, some aspect of wall motion during twisting and untwisting could be more effective as a stimulus than a simple increase in intraventricular pressure or change in diastolic volume. A small proportion of ventricular mechanoreceptors appear to be activated when the ventricular wall contracts forcefully around an almost empty chamber (Oberg & Thoren, 1972a). However, relatively few fibres, less than one-fifth, were found to be activated this way. Finally, it also appears that some ventricular afferents are bimodal, as these can be activated by both chemical and mechanical stimulation (Baker et al., 1979). Thus, any distinction between chemosensitive and mechanosensitive ventricular afferents is not absolute.

**Coronary artery afferents.** Many vagal branches terminate in the adventitia of the coronary arteries (Woollard, 1926). Brown (1965) recorded pulsatile afferent vagal activity related to coronary arterial pressure in anaesthetized cats, and suggested that this originated from mechanoreceptors in or near the coronary arteries. Notably, most of the fibres were active at rather low arterial pressure, which probably was related to contraction of surrounding myocardium. In anaesthetized dogs, vagal myelinated afferents with mechanoreceptors localized to the proximal coronary circulation were found to be activated by increases in coronary arterial perfusion (Drinkhill et al., 1993). Furthermore, the receptors were activated by ventricular contraction, but were much more sensitive to pressure changes directed to the coronary arteries than to increases in ventricular pressure. Probably, this explains why previous reports mistook coronary mechanoreceptors for ventricular receptors (Coleridge et al., 1964; Paintal, 1955; Thorén, 1977). Intracoronary injection of *Veratrum* alkaloids activated coronary afferent discharge in cats (Brown, 1965), but not dogs (Drinkhill et al., 1993), suggesting some species variation to chemical activation.

**Pulmonary artery afferents.** The existence of sensory nerve endings in the extrapulmonary portion of the pulmonary artery has been known for a long time (Nonidez, 1941; Whitteridge, 1948). The form and structure of these endings, which are distributed at the bifurcation and adjacent region of the main pulmonary artery and in proximal parts of the left and right braches, are consistent with a baroreceptor function (Bevan & Verity, 1962). Pulmonary arterial baroreceptors are supplied by myelinated afferent fibres (Coleridge et al., 1961). Furthermore, direct recordings from vagal afferent nerves in cats and dogs reveal a pattern of discharge synchronized with the rise of systolic pressure in the pulmonary artery (Bianconi & Green, 1959; Coleridge & Kidd, 1960; Pearce & Whitteridge, 1951). As is the case for aortic and carotid baroreceptors, discharge from pulmonary baroreceptors is more sensitive to the pulsatile nature of pressure, rather than mean pressure *per se* (Bevan & Kinnison, 1965; Coleridge & Kidd, 1961). Notably, sensitivity to pulsatility is augmented by intrathoracic pressure swings similar to that experienced during normal breathing (Moore et al., 2004b).

## Circulatory reflexes from the heart and the coronary and pulmonary arteries

**Laboratory animals.** A variety of approaches have been used to investigate reflexes from cardiopulmonary regions, including venous infusion, haemorrhage, obstruction of an inflow or outflow, distension of balloons, and changes in inotropic state. Use of such indiscriminate stimulation techniques, however, obscures how different receptors in different areas elicit different responses. In addition, experimental approaches that alter afferent transmission to study circulatory reflexes, such as cervical vagal cooling or anodal block, also are rather crude and non-selective. In the following section, we review evidence from studies that demonstrate that selective stimulation of discrete receptor populations in the heart and the coronary and pulmonary circulations elicits distinctive reflex responses.

*Atrial volume reflexes.* Distension of small balloons at the pulmonary vein–atrium junctions elicits reflex tachycardia in anaesthetized dogs (Ledsome & Linden, 1964). This effect, which is analogous to Bainbridge's reflex (Bainbridge, 1915), is mediated by an increase in activity in cardiac sympathetic efferent nerves without reciprocal reduction in cardiac vagal efferent activity (Linden et al. 1982a, 1982b). Furthermore, sympathetic activation affects only the sinoatrial node, since there is no accompanying change in cardiac inotropy (Furnival et al., 1971). Notably, reflex tachycardia induced by atrial receptor activation may vary with species. Boettcher et al. (1982) demonstrated that, despite comparable increases in cardiac filling pressures, heart rate rose significantly more during rapid infusion in dogs than baboons.

Reflex effects to discrete stimulation of atrial receptors have been studied in different vascular beds. Electrophysiological studies indicate that activation of atrial receptors inhibits neural activity in sympathetic efferent nerves directed to the kidney, whereas there is no significant effect on efferent activity to skeletal muscle or visceral abdominal areas (Karim et al., 1972). This sympathetic activity pattern is consistent with the observed vascular responses. For example, there is a reflex decrease in renal vascular resistance, whereas there is no consistent change in vascular resistance in the hind limb (Carswell et al., 1970a; Mason & Ledsome, 1974). In contrast, coronary vessels constrict, although only when heart rate and contractility are maintained (Drinkhill et al., 1989).

It has long been suggested that atrial receptors play a role in maintaining constant blood volume (Henry & Pearce, 1956; Henry et al., 1956). Stimulation of left atrial receptors results in an increase in urine flow and an increase of sodium excretion (Ledsome & Linden, 1968). Several factors may contribute to the renal component of atrial volume reflexes, such as suppression of renal sympathetic nerve activity and changes in renal haemodynamics. However, increased urine production also is evident in the denervated kidney (Ledsome et al., 1961), as well as in the isolated perfused kidney (Carswell et al., 1970b). Thus, some other, non-neural mechanisms also may be involved. Activation of atrial receptors leads to reduction in the release of antidiuretic hormone (Bennett et al., 1984), plasma renin (Drinkhill et al., 1988) and cortisol (Drinkhill & Mary, 1989), all of which contribute to control of urine flow.

The magnitude of reflex renal effects arising from atrial receptors may be influenced by blood volume and atrial size. Greater urine flow was observed in response to stimulation of atrial receptors in dogs with a high volume compared with dogs with a low volume (Gupta et al., 1982). Notably, systemic blood pressure, atrial pressure and heart rate were not different. In contrast to anaesthetized dogs, the renal effects of atrial receptor stimulation are less pronounced in anaesthetized monkeys (Gilmore & Zucker, 1978). Moreover, other studies suggest that volumetric control of renal excretion may involve different afferent and/or efferent mechanisms in non-human primates (Peterson et al., 1979, 1980).

Atrial receptors are ideally located to function as volume receptors, although the physiological importance in different species is controversial. Reflex tachycardia in some species will influence end diastolic volume. Moreover, regulation of urine formation and blood volume also will influence cardiac filling and diastolic volume. Thus, atrial receptor reflexes may maintain optimal diastolic filling and heart volume. This may be most prominent for quadrupeds, which maintain a horizontal position and do not experience orthostatically induced reductions in central blood volume. The relevance for humans, who spend most of the time upright or semi-upright, is less clear. Nevertheless, atrial volume reflexes may become important in humans exposed to prolonged bedrest. Furthermore, it is possible that atrial volume reflexes become dysfunctional under certain conditions, such as heart failure, leading to salt and water retention (Zucker et al., 1985).

*Ventricular receptor reflex.* Intracoronary injection of exogenous chemicals results in reflex bradycardia and vasodilatation of resistance vessels in skeletal muscle of anesthetized dogs (McGregor et al., 1986; Wright et al., 2000). This is the eponymous Bezold–Jarisch reflex (Jarisch & Zotterman, 1948; von Bezold & Hirt, 1867), also known as the coronary chemoreflex (Dawes & Comroe, 1954). Chemical mediators produced during myocardial ischaemia and reperfusion, such as bradykinin (Riccioppo Neto et al., 1974) and prostaglandins (Hintze & Kaley, 1984; Thames & Minisi, 1989), also elicit reflex bradycardia, vasodilatation, including in the kidneys,

and systemic hypotension. This may protect the heart and kidney by reducing myocardial oxygen demand and increasing renal blood flow.

Stimulation of ventricular mechanoreceptors, using a preparation that prevents the pressure stimulus from affecting other reflexogenic areas, induces modest reflex vasodilatation and bradycardia (Wright et al., 2000). This contrasts with considerably larger heart rate and vascular changes in response to intracoronary injection of veratridine to elicit the Bezold–Jarisch reflex. Furthermore, changing ventricular pressure over a physiological range had no effect on systemic vascular resistance. However, a small vascular response (<10%) was observed when ventricular pressure increased to above physiological levels (Drinkhill et al., 2001).

Changing ventricular inotropic state, by intracoronary injection of catecholamine or sympathetic nerve stimulation, elicited reflex vasodilatation in anaesthetized dogs (Emery et al., 1983; Fox et al., 1977). In these studies, however, like many others (Challenger et al., 1987; Tutt et al., 1988), no attempt was made to separate the stimulus to ventricular mechanoreceptors from that to coronary mechanoreceptors. Notably, when the pressure stimulus to the coronary circulation was kept constant, positive inotropic interventions did not induce a significant vascular response (al-Timman & Hainsworth, 1992; Drinkhill et al., 2001). Therefore, earlier reports of large responses arising from ventricular mechanoreceptors are almost certainly due to concomitant activation of coronary arterial baroreceptors. Furthermore, combination of sympathetic nerve stimulation with reduced ventricular filling was found to be ineffective as a stimulus for reflex vasodilatation (Drinkhill et al., 2001), which contradicts suggestions that forceful contraction of an inadequately filled chamber elicits a vasodepressor reflex (Thorén, 1979).

*Coronary baroreceptor reflex.* Early studies indicated a depressor reflex originating from the left coronary circulation (Brown, 1966; Kurihara, 1964). This was confirmed by Hainsworth and colleagues who performed a series of experiments using a complex extracorporeal perfusion circuit that enabled separate stimulation of coronary and ventricular mechanoreceptors (Fig. 2). As a result, increasing coronary arterial pressure elicits reflex dilatation in systemic vascular beds (al-Timman et al., 1993), which occurs via withdrawal of efferent sympathetic nerve activity to the vasculature, similar to that mediated by classical arterial baroreceptors (Drinkhill et al., 1996). Notably, renal sympathetic nerve activity is also reduced when coronary baroreceptors are activated. Unlike the classical baroreceptors, coronary mechanoreceptor activation does not elicit reflex bradycardia (Challenger et al., 1987; Tutt et al., 1988). Other notable characteristics of coronary baroreceptors include a much lower operating range than either aortic or carotid baroreceptors, and insensitivity to changes in arterial pulse pressure (McMahon et al., 1996b). Furthermore, coronary baroreceptors do not display signs of acute resetting (McMahon et al., 1998). Although the coronary and classical baroreflexes share a common efferent vascular pathway, recovery of vascular resistance takes much longer following unloading of coronary baroreceptors (McMahon et al., 1996a). This corresponds to a slow increase in efferent sympathetic nerve activity (Drinkhill et al., 1996). A delayed response is not apparent in the afferent limb of the reflex, suggesting that a central mechanism may be responsible for the prolonged inhibition of sympathetic efferent discharge. Thus, coronary arterial mechanoreceptors have the same vascular effect as classical arterial baroreceptors, although the contribution to beat-by-beat control of arterial pressure may be less pronounced due to lack of an effect on cardiac output. Notably, controlled stretch of the left anterior descending artery was found to elicit decreases in arterial pressure without changes in heart rate in conscious sheep (Bennetts et al., 2001). How input from coronary baroreceptors integrates with central cardiovascular control is not known. Nevertheless, coronary baroreceptors probably play a role in determining mean arterial pressure, revealed in the absence of functioning baroreceptors in denervated conscious dogs (Cowley et al., 1973; Daskalopoulos et al., 1984; Persson et al., 1988).

*Pulmonary arterial receptor reflex.* Early studies reported inconclusive findings for reflex vascular effects arising from attempted stimulation of pulmonary artery baroreceptors, using either inflation of a balloon in and around the bifurcation (Lewin et al., 1961; Osorio & Russek, 1962), or graded distension of the vascularly isolated and perfused pulmonary artery (Coleridge & Kidd, 1963). Subsequently, distension of the extrapulmonary portions of the pulmonary artery at rather higher static pressure was found to elicit reflex vasoconstriction and an increase in phrenic nerve activity (Kan et al., 1979; Ledsome & Kan, 1977; Ledsome et al., 1981; McMahon et al., 2000). Notably, the vascular response in skeletal muscle was quite large, but this was not accompanied by a change in renal vascular resistance (Ledsome & Kan, 1977). Furthermore, prolonged distension led to a sustained increased in systemic arterial pressure and a small increase in renal excretion of water and solutes, but only when pulmonary arterial pressure was raised above the physiological range (Kan & Ledsome, 1981). Eventually, in a preparation that combined pulsatile distension and intrathoracic pressure swings similar to those in an intact chest, it was established that pulmonary artery baroreceptors are sensitive over a physiological range and stimulate increases in sympathetic vasomotor

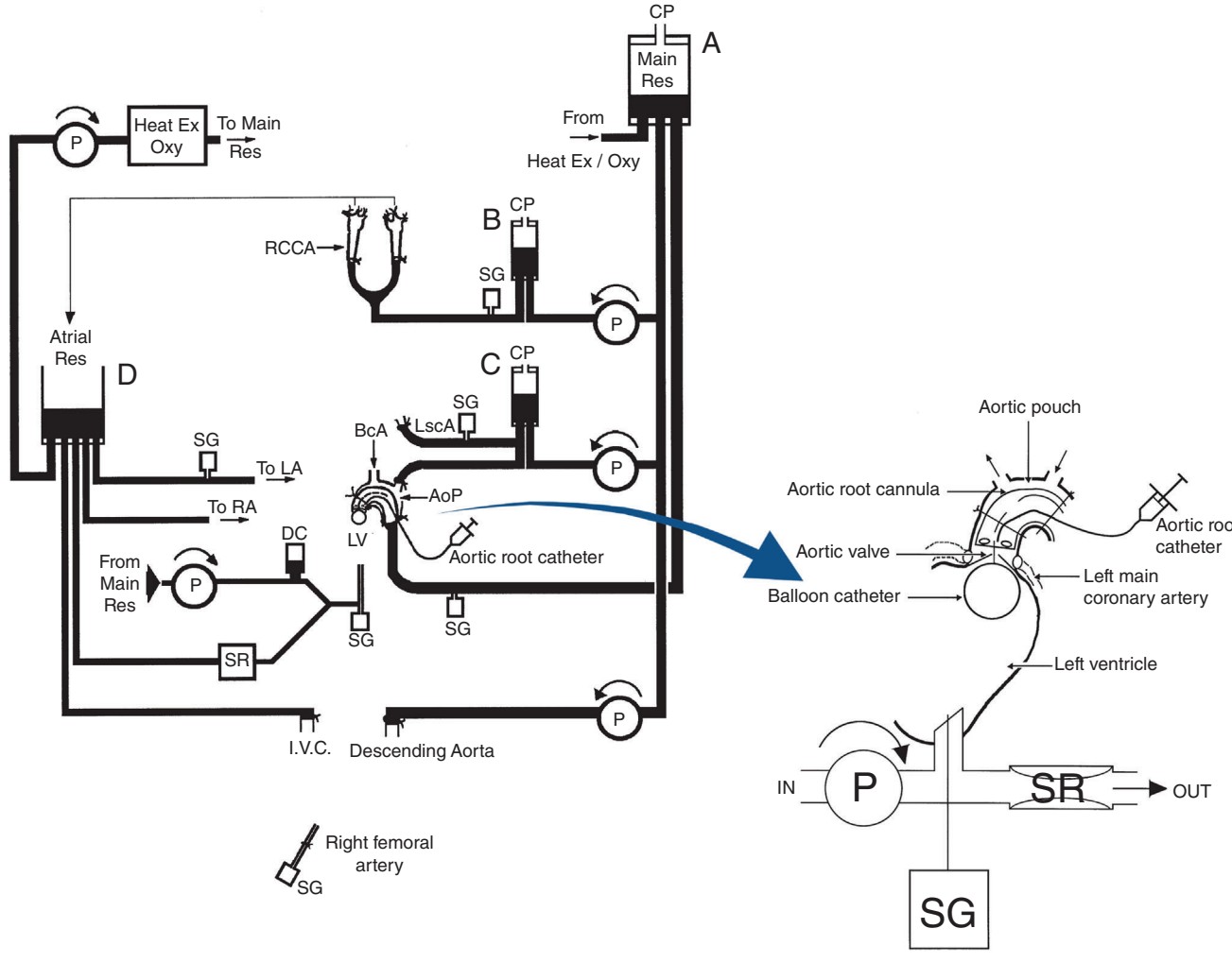

**Figure 2. Experimental model used to separate stimuli applied to aortic root, coronary arteries and left ventricle**

The model enables careful control of discrete reflexogenic sites in the heart, coronary circulation, aortic arch and carotid sinuses. A curved cannula introduced into the aortic arch, tied at the aortic root, distal to the origins of the coronary arteries and the left subclavian artery creates a pouch of aorta outside the cannula, and conveys blood to a main reservoir, A. Venous return to the heart is drained into an open reservoir, D, through cannulae tied into the inferior vena cava, and the left and right atria. Blood from D is pumped to A, via an oxygenator and heat exchanger. Blood from A is pumped to: (1) reservoir B, and at constant pressure into cannulae tied into the common carotid arteries; (2) reservoir C, and at constant pressure tied into the central and peripheral ends of the left subclavian artery; (3) the descending aorta at constant flow; and (4) the left ventricle (LV) through a damping chamber at constant flow and out through a Starling resistor to reservoir D. Cannulae inserted in both lingual arteries drain blood from the carotid bifurcation region to reservoir D. The LV is isolated from the coronary circulation by a balloon catheter inserted into the LV, which was positioned to occlude the aortic valve (enlarged inset). The balloon catheter was passed retrogradely across the aortic valve, inflated within the cavity of the left ventricle, and then repositioned to isolate the left ventricle from the coronary circulation. Insertion of an aortic root catheter, positioned to lie adjacent to the coronary ostia, provides a site for the intra-coronary injection of veratridine. AoP, aortic pouch; BcA, brachial cephalic artery; CP, constant pressure; DC, damping chamber; IVC, inferior vena cava; LscA, left subclavian artery; P, pump; RCCA, right common carotid artery; Res, reservoir; SR, Starling resistor; SG, strain gauge transducer. (Redrawn from Wright et al., 2001.)

tone and efferent activity recorded from renal sympathetic nerves (Moore et al., 2004a, 2011). Notably, whilst the threshold for vagal afferent excitation coincides with mean pulmonary arterial pressure around 12–15 mmHg, the threshold for vasoconstriction is around 20 mmHg. Moreover, the threshold for vasoconstriction is modulated by carotid sinus baroreceptor input, such that concurrent activation of carotid baroreceptors leads to an increase in the pulmonary artery pressure required to elicit systemic vasoconstriction.

Importantly, the reflex effects arising from low-pressure pulmonary baroreceptors are opposite to those of evoked by stimulation of the coronary, aortic and carotid baroreceptors. Furthermore, elevation of pulmonary arterial pressure modulates carotid baroreceptor function, resulting in rightward and upward resetting of the carotid-vasomotor baroreflex without any change in reflex gain (Moore et al., 2011). Such fundamental differences between pulmonary arterial baroreceptors and other baroreceptor reflexes often go unnoticed in the literature. Moreover, it is not clear how these contrasting effects integrate into normal beat-by-beat feedback control. However, an excitatory input arising from a sustained elevation of pulmonary arterial pressure is an intriguing possibility, and could couple exercise hyperpnoea (Wasserman et al., 1974) and baroreflex resetting (Melcher & Donald, 1981).

**Human studies.** Many techniques available for human study are unable to discriminate between reflexes arising from various cardiopulmonary reflexogenic areas. Nevertheless, Roddie et al. (1958) attributed reflex forearm vasodilatation in response to leg raising to engagement of mechanosensitive receptors in the intrathoracic vasculature. Sharpey-Schafer (1956) also observed similar changes in forearm blood flow during interventions that altered cardiac filling, but proposed that changes in pulsatility at the arterial baroreceptors triggered vascular responses. However, the case for intrathoracic receptors prevailed, giving rise to the concept of negative feedback control of the circulation by 'cardiopulmonary' mechanoreceptors.

*Integrated reflex circulatory control during mild hypovolaemia.* Application of lower body suction is a common non-invasive approach for study of reflex circulatory adjustments to reduced central venous pressure and cardiac filling. Moreover, it is assumed that low levels of lower body negative pressure (LBNP) enable 'cardiopulmonary' reflexogenic regions to be isolated from classical arterial baroreceptor reflex control of the circulation. Indeed, several studies report reductions in forearm blood flow and conductance (Abboud et al., 1979; Zoller et al., 1972), or increases in muscle sympathetic nerve activity (MSNA) (Pawelczyk et al., 2001; Vissing et al., 1989), without consistent changes in heart rate or arterial blood pressure. Notably, it is a lack of consistent change in mean arterial pressure that underpins the notion that the high pressure systemic arterial baroreceptors are not engaged by mild LBNP (Park et al., 2018).

Whether application of mild LBNP causes its effects through a change in the stimulus to 'cardiopulmonary' receptors or arterial baroreceptors has been debated. Mild levels of lower body suction have been shown to produce progressive decreases in cardiac output, followed subsequently by transient reductions in systolic and diastolic pressure that are restored within approximately 15 heartbeats (Fu et al., 2009). Logically, this correction in pressures can be attributed to negative feedback control by the systemic arterial baroreceptor reflex mechanism. A transient drop in stroke volume translates into reductions in pulse amplitude and pressure, which present as geometrical changes detected by ultrasound echo-tracking of carotid arterial bulb movements (Lacolley et al., 1992; Pannier et al., 1995), and magnetic resonance imaging measurements of cross-sectional area of the thoracic aorta (Taylor et al., 1995). A reduction in stroke volume and pulse pressure also probably explains elevated basal MSNA displayed by volume-depleted astronauts upon return to earth after spaceflight (Levine et al., 2002).

The significance of pulsatility in circulatory control is further highlighted by a study of patients implanted with continuous flow left ventricular assist devices (Cornwell et al., 2015). A reduction in pump speed led to increases in pulse pressure and pulsatile carotid arterial wall tension, which corresponded with reduced muscle sympathetic vasoconstrictor outflow despite a reduction in mean arterial pressure. Increasing the pump speed led to a reduction of pulse pressure and carotid arterial distortion and increased sympathetic vasomotor activity, which probably contributed to increased mean arterial pressure, although the effect of increased flow cannot be overlooked. Based on these findings, human baroreflex-mediated changes in sympathetic outflow are more sensitive to distension by pulse pressure than mean pressure. This is consistent with direct recordings of afferent activity in experimental animals, which indicate response patterns are more complex during pulsatile pressure (Chapleau & Abboud, 1987; Ead et al., 1952; James, 1971). Notably, a reduction in aortic baroreceptor activity was observed during non-hypotensive haemorrhage in anaesthetized dogs. Importantly, stroke volume and aortic pulse pressure declined steadily, and the reduced baroreceptor activity correlated closely with reductions in pulse pressure (Hakumäki et al., 1985). Furthermore, stroke volume is a major determinant of flow in baroreceptive arteries, with evidence that flow modulates baroreceptor activity in experimental animals (Hajduczok et al., 1988).

A study of heart transplant recipients observed reflex sympathetic activation and forearm vasoconstriction at low levels of LBNP despite cardiac denervation (Jacobsen et al., 1993). This contrasts with an earlier finding that forearm vasoconstriction during LBNP in was impaired after cardiac transplantation (Mohanty et al., 1987). Notably, these differences can be explained by a longer time elapsed between heart transplantation and experimental study (Jacobsen et al., 1993). Consequently, considering all the evidence, it appears than reduced mechanical distortion of systemic arterial baroreceptors, and not unloading of 'cardiopulmonary' receptors, primarily mediates reflex sympathetic activation and peripheral vasoconstriction during mild LBNP.

**Neurally mediated syncope.** Transient loss of consciousness, secondary to inadequate cerebral perfusion, is a relatively common medical problem. Furthermore, neurally mediated syncope has many triggers, including prolonged standing or sitting, hypovolaemia, overheating, emotional stress, pain, micturition, gastrointestinal stimulation and carotid sinus hypersensitivity. Conventional wisdom states that syncope occurs when normal autonomic control is interrupted by a sudden withdrawal of sympathetic tone and concomitant acute bradycardia. Hence, the term vasovagal is synonymous with neurally mediated syncope.

Similarity with the Bezold–Jarisch reflex in laboratory animals led to suggestions that activation of ventricular receptors may be a mechanism of neurally mediated syncope in humans (Abboud, 1989; Mark, 1983; Zucker, 1986). Indeed, alterations in the local biochemical environment may explain reflex bradycardia, systemic vasodilatation and hypotension observed during inferior posterior myocardial ischaemia and infarction (Chiladakis et al., 2003; Webb et al., 1972), or during coronary angiography (Eckberg et al., 1974; Perez-Gomez & Garcia-Aguado, 1977; Shah & Waxman, 2013). Moreover, a notion that exaggerated ventricular wall motion around an underfilled chamber acts as a trigger for fainting in humans also originates from observations in laboratory animals (Oberg & Thoren, 1972a). However, vasodepressor collapse is observed in people with transplanted, denervated hearts following infusion of a nitrovasodilator agent (Scherrer et al., 1990), and during upright tilt (Fitzpatrick et al., 1993). Furthermore, some echocardiographic studies do not support the 'underfilled heart' theory; it appears that the left ventricle is not necessarily empty, nor does ventricular contractility increase, at the point of syncope (Liu et al., 2000; Novak et al., 1996). Taken together, this evidence does not support stimulation of ventricular afferents as a trigger for neurally mediated syncope.

Fu and colleagues (2012) shed some light on pathways involved in neurally mediated syncope.

Two haemodynamic response patterns, based on the contribution of cardiac output and sympathetic vasoconstriction, were identified in healthy humans during upright tilt to presyncope. In around two-thirds of cases, a moderate fall in cardiac output was accompanied by a reduction of total peripheral resistance. In contrast, a second pattern consisted of marked fall in cardiac output, with no change in total peripheral resistance. The importance of this study is that it confirms that neurally mediated syncope is heterogeneous and can develop via vasodepressor or cardioinhibitory pathways. Furthermore, although considerable variability was observed in sympathetic neural responses, sympathetic withdrawal generally occurred late, after the onset of hypotension. This contrasts with earlier reports that sudden withdrawal of sympathetic activity precedes hypotension and neurally mediated syncope (Jardine et al., 2002; Morillo et al., 1997; Wallin & Sundlöf, 1982). Notably, persistence of sympathetic activity also was observed at the point of hypotension in healthy volunteers submitted to acute hypovolaemia (Cooke et al., 2009), as well as in fainters during head-up tilt (Vaddadi et al., 2010). Based on these findings, it seems that withdrawal of sympathetic vasoconstrictor tone and relaxation of vascular resistance are not preconditions of neurally mediated syncope. However, there may be disruption of the functional relationship between sympathetic neural activity and arterial blood pressure in postural syncope, regardless of whether sympathetic neural activity diminishes or not (Schwartz et al., 2013). Furthermore, firing properties and sympathetic neural recruitment patterns are more multifaceted than can be represented in integrated multi-unit MSNA neurograms. Analysis of the discharge behaviour of single-unit sympathetic discharge (Macefield et al., 1994), or recognition of action potential sub-populations within the raw MSNA signal (Salmanpour et al., 2010) should provide more insight into vascular sympathetic control during syncope.

*Neural control of the circulation during exercise.* Multiple control mechanisms evoke intensity-dependent changes in cardiac vagal outflow and sympathetic motor activity to the heart and blood vessels. Importantly, exercise-induced change in sympathetic vasomotor activity fulfils at least two functions. Vasoconstriction of inactive tissues enables redirection of cardiac output to regions with the greatest demand for flow, particularly contracting locomotor muscles. Furthermore, sympathetic restraint of the vast vasodilator capacity of contracting muscle is important to maintain the appropriate level of arterial pressure during upright dynamic exercise, even in elite athletes with very large stroke volume.

After some early uncertainty, it is known that human arterial baroreflex function is preserved during exercise, but it resets to operate around a higher exercising blood

pressure (Bevegård & Shepherd, 1966; Fadel et al., 2001; Potts et al., 1993). Notably, resetting of the cardiac and vasomotor components occurs without any change in reflex sensitivity. Exercise resetting, therefore, provides a regulatory pathway for integrating alterations in pressure, flow and resistance. However, physiological mechanisms for exercise resetting in humans are only partially understood. Central command, feed-forward impulses from cortical regions of the brain (Krogh & Lindhard, 1917; Ogoh et al., 2002; Williamson et al., 2002) and afferent neural feedback from exercising muscle (Alam & Smirk, 1937; Hureau et al., 2018; Papelier et al., 1997; Smith et al., 2003) appear to have the ability to independently reset the arterial baroreceptor reflex during static and dynamic exercise. Moreover, facilitatory interaction of signals from central command and afferent feedback appears to enhance baroreflex resetting (Iellamo et al., 1997).

It is proposed that afferent signals arising from 'cardiopulmonary' mechanoreceptors contribute to intensity-dependent resetting of the arterial baroreceptor reflex during exercise (Raven et al., 2019). In an upright position, transition from rest to exercise relocates blood that has pooled in the legs and increases cardiac filling pressure. It is suggested, therefore, that activation of 'cardiopulmonary' mechanoreceptors could explain why sympathetic vasoconstrictor nerve activity recorded from non-contracting muscle either decreases or does not change during mild leg-cycling exercise (Ichinose et al., 2008; Katayama et al., 2014; Notarius et al., 2015; Saito et al., 1993; Saito et al., 1997). Furthermore, it is proposed that 'cardiopulmonary' mechanoreceptor activation explains a downward shift in the operating point of the vasomotor component of arterial baroreceptor control during very mild cycling (Ichinose et al., 2008; Ogoh et al., 2003, 2007). However, it is suggested that the inhibitory effect of the 'cardiopulmonary' signal becomes attenuated by more powerful excitatory signals as exercise intensity increases, which enables intensity-dependent upward resetting of the vasomotor baroreflex and increases in MSNA (Katayama et al., 2020).

No attempt has been made to isolate a specific signal from cardiopulmonary reflexogenic regions during exercise in humans. Based upon available evidence, involvement of a signal originating from atrial receptors is doubtful. Activation of atrial volume receptors can elicit reflex tachycardia, although this appears to be species dependent (Boettcher et al., 1982). More importantly, atrial receptors do not regulate vasoconstrictor activity directed to skeletal muscle (Karim et al., 1972), which casts some doubt on a physiological role for these receptors as mediators of exercise-induced changes in MSNA. In addition, it is unlikely that a signal generated by more forceful ventricular contraction is involved: neither changing ventricular inotropic state nor loading the

left ventricular wall at high pressure has much effect on systemic vascular resistance in laboratory animals (Drinkhill et al., 2001; Wright et al., 2001). In our view, neural signals arising from two other cardiopulmonary reflexogenic regions, namely coronary artery and pulmonary artery baroreceptors, are most likely to contribute to exercise resetting across the intensity spectrum. Increasing perfusion pressure in the coronary circulation, particularly during transition to low- to mild-intensity exercise, could explain reports of reduced sympathetic outflow to skeletal muscle (Katayama et al., 2014; Saito et al., 1997). On the other hand, intensity-dependent elevation of pulmonary artery pressure, which can be by 40–50% during mild exercise and 85% at maximal intensity (Kovacs et al., 2009; Wright et al., 2016), could activate a neural signal that contributes to upward resetting and greater sympathetic restraint of exercise hyperaemia in parallel with central command and muscle afferent feedback. Moreover, systolic pulmonary arterial pressure increases, by approximately 15–20% from rest, during static lower limb exercise and when blood flow to the limb is restricted (White et al., 2013). Taken together, a neural signal from the pulmonary artery could provide an additional functional pathway for exercise resetting and sympathetic restraint. Therefore, we suggest a conceptual framework for exercise resetting which considers separate feedback signals from coronary and pulmonary arterial baroreceptors (Fig. 3). Notably, neither coronary nor pulmonary arterial baroreceptors influence neural control of the heart in laboratory animals, so any involvement in mediating exercise resetting would be confined to the vasomotor component of blood pressure regulation. Clearly, rigorous mechanistic testing is required to delineate the potential roles of coronary and pulmonary arterial baroreceptors in relation to central command and skeletal muscle afferent feedback.

*Responses to arterial pressure changes in human coronary and pulmonary circulations.* Very few studies have attempted to isolate stimuli to coronary and pulmonary arterial systems in humans. However, in a small cohort of patients undergoing mitral valve surgery, cardiopulmonary bypass enabled study of vascular responses to step changes in coronary perfusion pressure (Kincaid et al., 2005). A small reduction in systemic vascular resistance was observed in response to a step increase in coronary pressure, whereas responses to a drop in coronary pressure were variable. The importance of these findings, albeit in patients under general anaesthesia, is that they support the evidence from experimental animals and suggest the presence of coronary baroreceptors in humans.

To assess the potential of a neural input arising from pressure elevation in the pulmonary artery, Simpson et al. (2020) measured MSNA in healthy lowland natives

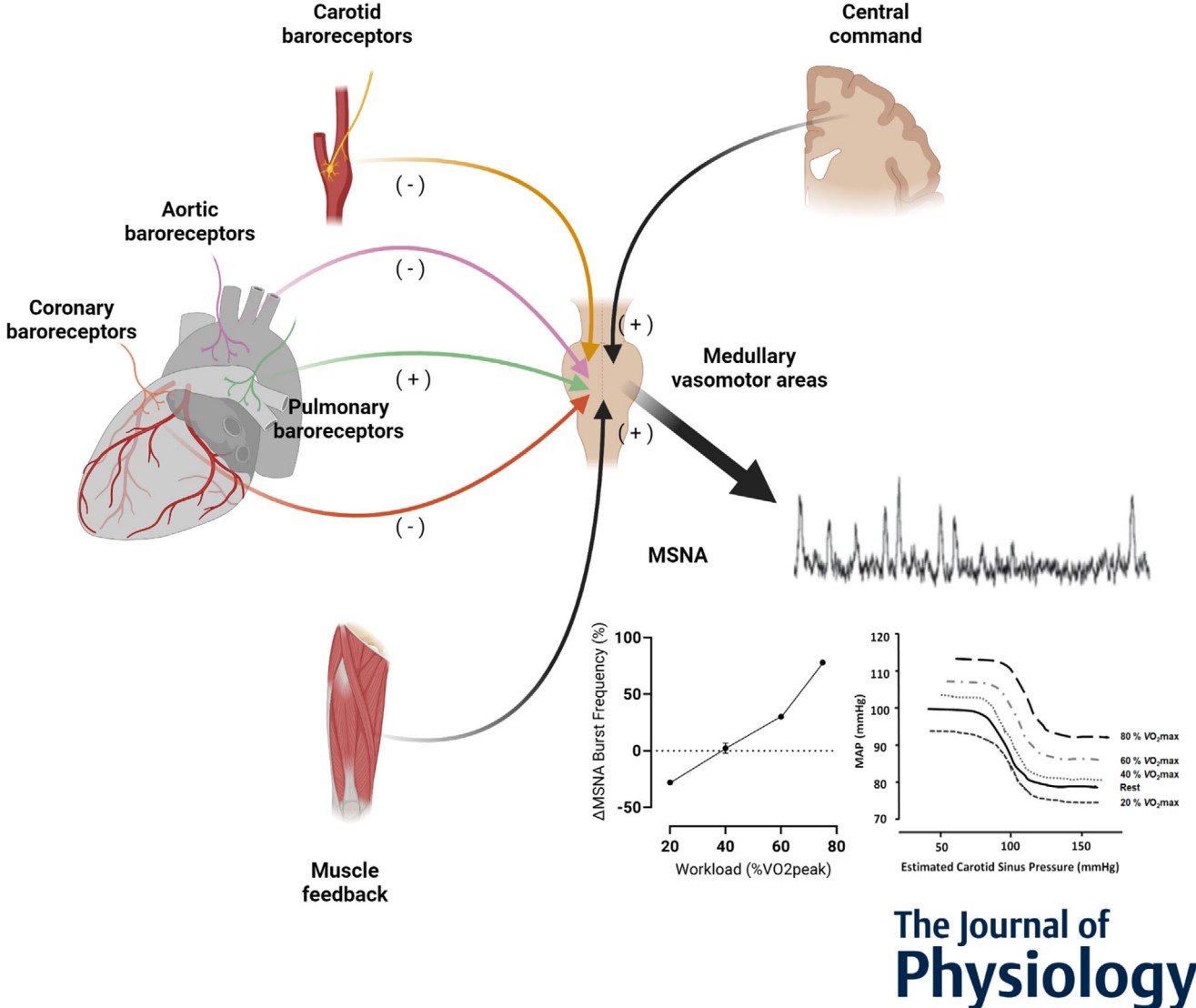

**Figure 3. A schematic representation of the conceptual framework for neural mechanisms mediating vasomotor baroreflex resetting and changes in vasoconstrictor outflow directed to skeletal muscle during exercise**

Neural signals arising from the brain (central command) and afferent input from carotid and aortic baroreceptors, coronary baroreceptors, pulmonary arterial baroreceptors, and skeletal muscle mechano- and metaboreceptors converge centrally within vasomotor control areas in the medulla oblongata. (Adapted from Fadel & Raven (2012).) The influence of each signal on sympathetic vasoconstrictor output (MSNA) varies in the transition from rest to exercise and with progressively increasing intensity during dynamic exercise. This is summarized in the line graph. (Adapted from Katayama & Saito (2019).) A reduction in sympathetic vasomotor outflow ($\Delta$MSNA%) is apparent during light (20% $\dot{V}_{O_2max}$) intensity. As intensity increases, MSNA gradually rises in proportion to workload (% $\dot{V}_{O_2max}$). Also shown are the carotid-MAP (vasomotor) stimulus responses curves at rest and during incremental exercise (20–80% $\dot{V}_{O_2max}$). Lines represent a logistic function model fitted to mean data (not shown). The MAP response curve at 20% $\dot{V}_{O_2max}$ is reset downward, whereas the curves are reset upward and rightward during higher intensity exercise. There is no change in sensitivity at any exercise intensity. (Adapted from Ogoh et al. (2003).) In this framework, the mechanism for inhibition of MSNA and downward resetting at low intensity exercise is loading of coronary arterial baroreceptors, despite central command. As exercise progresses and intensity increases, central command escalates and excitatory inputs from skeletal muscle receptors and pulmonary arterial baroreceptors integrate to reset the vasomotor baroreflex and activate sympathetic vasoconstrictor neurones.

exposed to high altitude hypoxia. Drawing on evidence from animal experiments, this study tested the hypothesis that an acute reduction in pulmonary vascular resistance and arterial pressure, mediated by inhalation of nitric oxide, would elicit a reduction in sympathetic vasoconstrictor activity and reset the sympathetic baroreflex. In fact, a 20% reduction in systolic pulmonary arterial pressure elicited a 20% reduction in muscle sympathetic outflow, which was accompanied by a reduction in the operating point of the sympathetic vasomotor baroreflex. This suggests that elevated pulmonary artery pressure contributes to reflex sympathoexcitation and vasomotor baroreflex resetting, in the case of high altitude to counter hypoxic vasodilatation and prevent situational hypotension (Simpson et al., 2019). Involvement of this mechanism also could explain the observation that breathing 100% oxygen, to manipulate hypoxic chemoreceptor drive, has little effect on basal sympathetic outflow and vascular sympathetic baroreflex function in lowlanders and high-altitude natives (Simpson et al., 2019; Simpson et al., 2021). Such findings are consistent with a notion that peripheral chemoreflex activation is not solely responsible for sympathetic neural overactivity during prolonged exposure (Fisher et al., 2018; Hansen & Sander, 2003). Whilst sensitization of carotid bodies and central amplification of chemoreceptor afferent input cannot be ruled out as a mechanism, recent human studies indicate dissociation of ventilatory and sympathetic responses to peripheral chemoreflex activation during acute hypoxia (Keir et al., 2019; Prasad et al., 2020). If differential control persists during prolonged hypoxia, there could be uncoupling of ventilatory and sympathetic acclimatization to high altitude, which raises the possibility that multiple mechanisms account for sympathetic acclimatization to chronic hypoxia. This could explain continuation of heightened sympathetic activity following return to sea level (Hansen & Sander, 2003; Mitchell et al., 2018), which may, in part, be attributed to incomplete reversal of high-altitude pulmonary resistance and elevated pulmonary arterial pressure (Groves et al., 1987; Maufrais et al., 2017).

Neural input from pulmonary arterial baroreceptors also could explain so-called paradoxical excitation of single unit muscle sympathetic vasoconstrictor fibres, observed during acute elevation of right heart and pulmonary artery pressures (Millar et al., 2013, 2015; Incognito et al., 2019). In contrast, multiunit MSNA was reduced when pulmonary artery pressure increased secondary to rapid saline infusion, (Pawelczyk et al., 2001), although this may reflect activation of systemic baroreceptors, by widening of the arterial pressure pulse, which was more influential than activation of pulmonary artery baroreceptors. This juxtaposition also could reflect differences in granularity of information contained within single-unit and multiunit neurograms. Hence,

the contribution of pulmonary artery mechanoreceptors to feedback control of sympathetic outflow in a closed loop is uncertain. It is possible that the contribution of pulmonary artery baroreceptors to sympathetic vasomotor control is revealed primarily during large-step rather than incremental change, when the excitatory signal exceeds other counteracting inputs and/or contributes to vasomotor baroreflex resetting. Of clinical importance, a persistent sympathoexcitatory effect arising from pulmonary artery baroreceptors may contribute to myriad mechanisms driving sympathetic overactivity in pathophysiological states including heart failure (Ferguson et al., 1990), pulmonary arterial hypertension (Velez-Roa et al., 2004) and sleep apnoea (Somers et al., 1995). Notably, none of the strands of evidence discussed here represent proof of a role for pulmonary artery mechanoreceptors in human sympathetic vasomotor output. However, the contribution of this afferent input to sympthoexcitation warrants further consideration.

## Summary

Neural control of blood pressure, the regulated variable of the cardiovascular system, involves integration of multiple inputs to the central nervous system. Systemic arterial baroreceptor reflexes are very effective at stabilizing acute changes in arterial blood pressure. However, inputs from other reflexogenic areas also have important roles. We have highlighted some key differences in reflex effects originating from discrete populations of mechanoreceptors in the walls of the atria, and the coronary and pulmonary arteries. Atrial receptors signal normal atrial wall mechanical events and contribute, primarily, to control of heart rate and intravascular volume, although this may be more important in quadrupeds than humans and other primates. There is no strong evidence that atrial receptors exert any reflex control over arterial pressure via changes in systemic vascular resistance. Coronary baroreceptors signal normal arterial pressure changes in the coronary circulation and exert an inhibitory effect over vascular tone like that from systemic arterial baroreceptors. This occurs without any noticeable control over heart rate. Coronary receptors operate over a small range of perfusion pressure, and are equally sensitive to static and pulsatile pressure. Unloading of coronary baroreceptors is followed by slow vasoconstriction and recovery of sympathetic efferent activity. Whereas carotid and aortic baroreceptors are involved in rapid beat-to-beat control, it is possible that coronary baroreceptors may be more concerned with regulation of mean blood pressure over longer periods. Pulmonary artery baroreceptors are unique among mechanosensitive receptors attached to vagal myelinated afferents. Like other groups, pulmonary arterial baroreceptors are tonically active, but they exert

an excitatory influence over systemic vascular tone when stimulated. Input from pulmonary baroreceptors may contribute to baroreflex resetting and sympathetic restraint of muscle blood flow during exercise and under conditions of prolonged systemic hypoxia. A fourth group of receptors, located predominantly in the left ventricle, stand apart from the other three groups. Ventricular receptors are attached to non-myelinated afferents and appear to be relatively insensitive to ventricular wall movements, although overdistension does elicit a small depressor effect. It is probable that a large depressor response attributed to mechanical distortion of these receptors actually originates in the coronary baroreceptors. On the other hand, activation of ventricular receptors by naturally occurring chemicals elicits a profound depressor response, which may have some protective function during ischaemic myocardial injury. In summary, neural circulatory control originating from cardiopulmonary reflexogenic regions is complex, having more aspects than is sometimes depicted. We encourage future research that focuses on separating the unique contributions of atrial, coronary and pulmonary arterial baroreceptors, thus enabling more complete understanding of reflex control of the circulation.

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

## Additional information

### Competing interests

None.

### Author contributions

JPM. was responsible for the concept of the article and wrote the first draft of the manuscript. All authors contributed to the analysis and interpretation of studies reviewed in the manuscript, critically evaluated the manuscript for important intellectual content and revised edited subsequent drafts. All authors approved the manuscript and agreed to be accountable for all aspects of the work in ensuring that questions related to the accuracy or integrity of any part of the work are appropriately investigated and resolved. All persons designated as authors qualify for authorship, and all those who qualify for authorship are listed.

### Funding

Research at Leeds University was funded through a series of grants awarded by the Medical Research Council and British Heart Foundation.

### Acknowledgements

MJD and JPM are grateful to former colleagues in 'cardiovascular studies' at Leeds University, particularly Prof Roger Hainsworth and Dr David Mary.

### Keywords

baroreceptor reflex, cardiovascular control, sympathetic nerve activity, vagal afferent

### Supporting information

Additional supporting information can be found online in the Supporting Information section at the end of the HTML view of the article. Supporting information files available:

**Peer Review History**

