## [Peer Review History · The Journal of Physiology]

DIFFERENTIAL CONTRIBUTIONS OF CARDIAC, CORONARY AND PULMONARY ARTERY VAGAL MECHANORECEPTORS TO REFLEX CONTROL OF THE CIRCULATION

Jonathan P Moore, Lydia L Simpson, and Mark J. Drinkhill
DOI: 10.1113/JP282305

Corresponding author(s): Jonathan Moore (j.p.moore@bangor.ac.uk)

The following individual(s) involved in review of this submission have agreed to reveal their identity: Irving H. Zucker (Referee #1); Benjamin D Levine (Referee #2)

Review Timeline:

Submission Date:	18-Oct-2021
Editorial Decision:	21-Dec-2021
Revision Received:	11-Mar-2022
Editorial Decision:	04-May-2022
Revision Received:	27-Jun-2022
Accepted:	19-Jul-2022

Senior Editor: Ian Forsythe

Reviewing Editor: Emma Hart

Transaction Report:

Dear Dr Moore,

Re: JP-TR-2021-282305 "Time to retire the "cardiopulmonary" baroreflex and focus on differentiated vascular control from cardiac, coronary, and pulmonary arterial receptors" by Jonathan P Moore, Lydia L Simpson, and Mark J. Drinkhill

Thank you for submitting your Topical Review to The Journal of Physiology. It has been assessed by a Reviewing Editor and by 2 expert referees and I pleased to tell you that it is considered to be acceptable for publication following satisfactory revision.

The reports are copied at the end of this email. Please address all of the points and incorporate all requested revisions, or explain in your Response to Referees why a change has not been made.

NEW POLICY: In order to improve the transparency of its peer review process The Journal of Physiology publishes online as supporting information the peer review history of all articles accepted for publication. Readers will have access to decision letters, including all Editors' comments and referee reports, for each version of the manuscript and any author responses to peer review comments. Referees can decide whether or not they wish to be named on the peer review history document.

I hope you will find the comments helpful and have no difficulty in revising your manuscript within 4 weeks.

Your revised manuscript should be submitted online using the links in Author Tasks Link Not Available. This link is to the Corresponding Author's own account, if this will cause any problems when submitting the revised version please contact us.

You should upload:

- A Word file of the complete text (including any Tables);
- An Abstract Figure, (with accompanying Legend in the article file)
- Each figure as a separate, high quality, file;
- A full Response to Referees;
- A copy of the manuscript with the changes highlighted.
- Author profile. A short biography (no more than 100 words for one author or 150 words in total for two authors) and a portrait photograph of the two leading authors on the paper. These should be uploaded, clearly labelled, with the manuscript submission. Any standard image format for the photograph is acceptable, but the resolution should be at least 300 dpi and preferably more.

- A 'Cover Art' file for consideration as the Issue's cover image;
- Appropriate Supporting Information (Video, audio or data set https://jp.msubmit.net/cgi-bin/main.plex?form_type=display_requirements#supp).

To create your 'Response to Referees' copy all the reports, including any comments from the Senior and Reviewing Editors into a Word, or similar, file and respond to each point in colour or CAPITALS. Upload this when you submit your revision.

I look forward to receiving your revised submission.

Yours sincerely,

Ian D. Forsythe
Deputy Editor-in-Chief
The Journal of Physiology
<https://jp.msubmit.net>
<http://jp.physoc.org>
The Physiological Society
Hodgkin Huxley House
30 Farringdon Lane
London, EC1R 3AW
UK
<http://www.physoc.org>
<http://journals.physoc.org>

EDITOR COMMENTS

Reviewing Editor:

Thank you for submitting this review to the Journal of Physiology. The review is well written and I enjoyed reading it. There are however several issues that need to be improved on; some of which have also been highlighted by both reviewers. These include:

1. In parts the review is not very comprehensive. Key evidence and studies are not cited, some of which do not support the argument for a physiological role of the cardiac and pulmonary receptors. These need to be included. See studies mentioned by both reviewers. Also, the authors do rely on citing review articles. Please cite the original evidence to help support some of your arguments.
2. Although mostly well written, I did feel like some sections were not 100% clear and I think this is because the authors rely on using a systemic response or lack of response as evidence the reflexes/receptors are important/not important; but this is usually when only one part of the circulation is monitored. Reflex responses to changes in , for example, ventricular pressure/distension/volume might be much more selective versus that for arterial baroreflexes. For example perhaps vasoconstriction in renal beds/splanchnic beds happens without any change in systemic vascular resistance. I think this should be pointed out by the authors, or say why responses in only one part of the circulation are being focused on.
3. In the ventricular reflex section; the authors mention increases in pressure but seem to miss decreases in ventricular pressure (e.g. what would happen during arrhythmia when the ventricles might empty before they have filled?). Also can the authors mention the role of mechanical distortion without changes in pressure? (i.e. mechanical twist)
4. Vasovagal syncope - there are multiple causes of vasovagal syncope, potentially cardiac/pulmonary receptors have a role in some instances, but the majority of vasovagal syncope seems to be driven by psychogenic factors.
5. I think the authors need to point out the limitations of assuming a causative role of elevated pulmonary arterial pressure in driving increased SNA during hypoxia, when using correlations.
6. Title: This needs to be shortened. Also perhaps it is a bit misleading - not everyone thinks of the cardiopulmonary baroreflex as one reflex. This is highlighted by JR Levick's textbook where afferent neurones and their role in the coronary vessels, atria/ventricles and pulmonary arteries are reviewed. (so it isn't always textbook dogma).

Senior Editor:

Thanks for an interesting review. Please consider the important points raised by the referees and RE in their review. I wonder if you should add an additional figure which explains the background for the cardiopulmonary baroreflexes and helps introduce your topic to a wider audience.

REFEREE COMMENTS

Referee #1:

The authors provide a brief overview of reflexogenic areas of the heart and great vessels with the intent of dispelling the idea that there are cardiopulmonary stretch reflexes that play a homeostatic role in either blood pressure or blood volume regulation. In large part, they argue that these receptors and their reflexes (especially in the ventricles) operate only under extreme circumstances and thus are not regulatory. While this reviewer believes that the evidence for selective loading and unloading of cardiopulmonary receptors of vagal origin is not conclusive and there is much room for speculation, the paper itself is superficial in nature and does not really represent a comprehensive review. There are several studies that could have been added. Many review articles are cited rather than original works. Furthermore, the discussion concerning the role of atrial receptor activation may be true for acute effects but ignores the potential for chronic regulation. Finally, there is no discussion of cardiac afferents of spinal origin. Specific comments are below:

Page 4, line 45: Please provide references for this statement.

Page 7, line 126: Figure 2 should be figure 1.

Page 9, para 2: Here one should discuss the paper by Boettcher et al (AJP, 1982 PMID: 7065218) where it was shown that the Bainbridge reflex was markedly blunted in humans and baboons. In the last paragraph on page 9 and beginning of page 10 it might be useful to discuss two papers by Peterson et al. on volume expansion in primates (PMID:6767667 and 111262).

Page 11, para 1: The point is well taken here but the paper of Hakumaki and Goetz should be cited (PMID: 3927749).

Page 11: line 240: I would eliminate the word "overwhelming". This is editorial license and in this reviewer's opinion, the

evidence is not overwhelming.

Page 12: Dynamic exercise. I think this section should be expanded. Muscle afferent feedback in the regulation of sympathetic outflow generally occurs during static exercise or when blood flow to the limb is restricted.

Page 13, line 278: Figure 3 should be figure 2.

Page 14, line 327: This may be true but effects on renal resistance and renal sympathetic nerve activity may be an important role of atrial receptor activation, especially chronically.

Referee #2:

I enjoyed reading the paper by Moore et al which is a well written review of the multiplicity of circulatory afferents involved in cardiovascular control. It is scholarly and thorough, and very informative merging historical data with more recent insights. I have a few suggestions that might improve the review that I offer to the authors for consideration.

1). The authors set up the idea of "cardiopulmonary" baroreflexes as a straw man that they plan to knock down, and I understand that in the current climate of AltMetrics and social media, coming up with a provocative title might enhance downloads and attention. And if this paper was an editorial or "on my mind" type of piece, then I would be more accepting of that approach. However it is a review paper in the premier physiology journal, and the majority of the paper appropriately presents a thoughtful integration of science rather than a debate. Therefore I suggest that the authors reframe the paper and change the title to avoid sensationalism and focus on the scholarly nature of their presentation.

2). The authors put forth the hypothesis that pulmonary artery baroreceptors contribute to sympathetic activation during exercise and hypoxia. However I am not convinced that these receptors are physiologically meaningful, particularly in relation to other well known inputs, such as skeletal muscle afferents and chemoreceptors. PA pressure does not rise during small muscle mass handgrip exercise, yet it leads to marked sympathetic activation, even during circulatory arrest when there is no muscle contraction. Moreover volume infusion which clearly raises PA pressure is associated with sympathetic silence, as shown by Jim Pawelczyk in the authors cited reference from 2001. I also am not clear how the authors are separating feedback from pulmonary baroreceptors from chemoreflexes, at least quantitatively. I would be very cautious about interpreting the association between PAP and MSNA during hypoxia, when both stimuli are changing simultaneously as representing cause and effect. As only one example, there is a near perfect relationship between max VE in L/min and maxVO₂ in L/min - not because VE determines VO₂, but because both vary with body size. I suggest the authors revisit this formulation, and try to put the role of pulmonary baroreceptors in context.

3). I think the authors should also consider that ventricular mechanoreceptors are not necessarily sensitive only to ventricular pressure. So as the heart twists and translates as it contracts, INTRAMYOCARDIAL pressure can go up quite high, clearly well above systolic arterial pressure. So I am not sure that the "normal physiological range" for these receptors is below the activating pressures for ventricular mechanoreceptors. It is also well known that common stimuli like ischemia, or injection of contrast dye will stimulate ventricular afferents, not just external chemical irritants, so these receptors are often responsible for hypotension during myocardial infarction, or coronary angiography. Please see work by Ron Victor who did a variety of studies in the cath lab examining responses in different vascular beds in innervated and denervated hearts to injection of contrast dye.

4). The authors use a lot of space to distinguish between myelinated and unmyelinated fibers. I understand why they do so, but it is never stated explicitly, so might be confusing, especially to a clinically oriented reader.

5). I think the authors neglect some important work supporting the idea that it is pulsatile distortion or arterial baroreceptors by SV that play a critical role in regulation of sympathetic activity during changes in cardiac filling. This is true after spaceflight where both plasma volume and ventricular remodeling occur, and also in patients with continuous flow left ventricular assist devices. The paper by Cornwell et al (Circulation. 2015 Dec 15;132(24):2316-22) is remarkable in that it shows that an intervention that INCREASES mean arterial pressure (and also coronary perfusion pressure) but reduces pulsatility leads to INCREASES in MSNA. I'd like to learn the authors thoughts on how these studies inform their conceptual framework

6). I think the authors are too dismissive of the idea that sympathetic withdrawal occurs during some patients with syncope. As noted above, even if the heart is not empty, there is often marked increases in contractility during hypovolemia which alters intramyocardial pressure even if the walls of the heart are not smacking against each other. Fu et al (J Physiol. 2012 Apr 15;590(8):1839-48) have published a variety of different pathways that syncope may occur in healthy individuals, including those with and without sympathetic withdrawal. There are multiple pathways by which this kind of neurally mediated syncope can occur.

7). a few specific points

- Page 3, line 6: what "unphysiological" stimuli are you talking about here?

- Page 9, line 185 - how do atrial receptors regulate "heart size"? Do you mean central blood volume? Atrial or ventricular filling?

REQUIRED ITEMS:

-Author profile(s) must be uploaded via the submission form. Authors should submit a short biography (no more than 100 words for one author or 150 words in total for two authors) and a portrait photograph of the two leading authors on the paper. These should be uploaded, clearly labelled, with the manuscript submission. Any standard image format for the photograph is acceptable, but the resolution should be at least 300 dpi and preferably more. A group photograph of all authors is also acceptable, providing the biography for the whole group does not exceed 150 words.

-It is the authors' responsibility to obtain any necessary permissions to reproduce previously published material
https://jp.msubmit.net/cgi-bin/main.plex?form_type=display_requirements#use

END OF COMMENTS

Confidential Review

18-Oct-2021

AUTHOR RESPONSE

Thank you for the positive responses to our review. We are very pleased to have this opportunity to respond to Editor and Reviewer Comments and to submit a revised version of our review.

EDITOR COMMENTS

Reviewing Editor:

Thank you for submitting this review to the Journal of Physiology. The review is well written and I enjoyed reading it. There are however several issues that need to be improved on; some of which have also been highlighted by both reviewers. These include:

1. In parts the review is not very comprehensive. Key evidence and studies are not cited, some of which do not support the argument for a physiological role of the cardiac and pulmonary receptors. These need to be included. See studies mentioned by both reviewers. Also, the authors do rely on citing review articles. Please cite the original evidence to help support some of your arguments.

We agree our original submission lacked some detail. Given the long history of studies, it was a challenge to cover all aspects comprehensively within the suggested 'word count'. To address this, and other Reviewer comments, we have added additional key evidence and studies, some of which were identified by the Reviewers. Furthermore, we have also cited more primary sources. Hence, this revised article is considerably longer than the original submission.

2. Although mostly well written, I did feel like some sections were not 100% clear and I think this is because the authors rely on using a systemic response or lack of response as evidence the reflexes/receptors are important/not important; but this is usually when only one part of the circulation is monitored. Reflex responses to changes in, for example, ventricular pressure/distension/volume might be much more selective versus than for arterial baroreflexes. For example perhaps vasoconstriction in renal beds/splanchnic beds happens without any change in systemic vascular resistance. I think this should be pointed out by the authors, or say why responses in only one part of the circulation are being focused on.

Thank you for this comment. We recognize the Reviewing Editor's concern and have tried to highlight the selective nature of the responses in the different parts of the circulation. To make this revision clearer, we highlight that cardiovascular receptors in the heart and pulmonary arteries are depicted, typically, as having a *net* tonic inhibitory effect, similar to arterial baroreceptors. However, in our experience, this is not the case. We provide more details about atrial and ventricular receptor reflexes. We have included more evidence of renal effects, when appropriate, which may occur independently of reflex changes in systemic vascular resistance.

3. In the ventricular reflex section; the authors mention increases in pressure but seem to miss decreases in ventricular pressure (e.g. what would happen during arrhythmia when the ventricles might empty before they have filled?). Also can the authors mention the role of mechanical distortion without changes in pressure? (i.e. mechanical twist).

Thank you for the comment. We have now included a section discussing the possibility that twisting and untwisting of the myocardium might be more effective as a stimulus than a simple change intraventricular pressure or change in diastolic volume (Pg. 5-6, Ln 78-80). However, we cannot speak to how different intramyocardial tension at the same ventricular end diastolic and peak systolic pressures might interact, due to a lack of research to support this. It is plausible that

curvature of the ventricle might influence receptors according to their location, although most receptors appear to be localized to the inferior posterior wall.

We are not quite clear on the question regarding the effect of arrhythmia. If we understand correctly, the reviewing editor is asking if lower than optimal ventricular volume, for example during a premature ventricular contraction, would result in the Bezold-Jarisch reflex. In our experience, reduced LV peak pressure and LVEDP has no effect on systemic vascular resistance or heart rate when combined with increased contractility. Furthermore, injection of veratridine under these conditions elicited a classic Bezold- Jarisch response (Drinkhill et al, 2001).

4. Vasovagal syncope - there are multiple causes of vasovagal syncope, potentially cardiac/pulmonary receptors have a role in some instances, but the majority of vasovagal syncope seems to be driven by psychogenic factors.

Thank you for this comment. We have added to the section to address this ((Pg 14-15, ln 320- 322), and addressed other points raised by Reviewer 2.

5. I think the authors need to point out the limitations of assuming a causative role of elevated pulmonary arterial pressure in driving increased SNA during hypoxia, when using correlations.

Thank you for this comment. The purpose of our study (Simpson et al. 2020) was to recapitulate evidence from laboratory animals, albeit under differential experimental conditions (i.e., high altitude hypoxia). That the findings of our study are compatible with those of correlational research is a useful observation, but this was not the *raison d'être* of our study. We have revised this section of the review to reflect this (Pg 19 ln 440).

6. Title: This needs to be shortened. Also perhaps it is a bit misleading - not everyone thinks of the cardiopulmonary baroreflex as one reflex. This is highlighted by JR Levick's textbook where afferent neurones and their role in the coronary vessels, atria/ventricles and pulmonary arteries are reviewed. (so, it isn't always textbook dogma).

We accept that the title is misleading, and we have adjusted it. Furthermore, to provide a more nuanced assessment of textbook knowledge, we have reframed some elements of this review (*per* Reviewer 2, comment 1). Fundamentally, our intention is to remind/inform the reader that cardiac atrial, coronary and pulmonary artery receptors are not all inhibitory in nature, which is not always reflected in textbooks.

Senior Editor:

Thanks for an interesting review. Please consider the important points raised by the referees and RE in their review. I wonder if you should add an additional figure which explains the background for the cardiopulmonary baroreflexes and helps introduce your topic to a wider audience.

We have added a figure from Levick's Introduction to Cardiovascular Physiology 6th Edition. Hopefully we will be able to obtain the publishers permission to use this, or a version of it.

REFEREE COMMENTS

Referee #1:

The authors provide a brief overview of reflexogenic areas of the heart and great vessels with the intent of dispelling the idea that there are cardiopulmonary stretch reflexes that play a homeostatic role in either blood pressure or blood volume regulation. In large part, they argue that these receptors and their reflexes (especially in the ventricles) operate only under extreme circumstances and thus are not regulatory. While this reviewer believes that the evidence for selective loading and unloading of cardiopulmonary receptors of vagal origin is not conclusive and there is much room for speculation, the paper itself is superficial in nature and does not really represent a comprehensive review. There are several studies that could have been added. Many review articles are cited rather than original works. Furthermore, the discussion concerning the role of atrial receptor activation may be true for acute effects but ignores the potential for chronic regulation. Finally, there is no discussion of cardiac afferents of spinal origin.

Thank you for your helpful comments. We were constrained by a word limit, but we accept your concerns surrounding the breadth and depth of the first submission. To address these, we have included the studies you suggested, and some additional studies to provide a more comprehensive review of the literature. In addition, we have extended our section on atrial volume receptors to address their potential importance in chronic regulation. Also, we have sought to provide more detail to underpin our view of vagal ventricular reflexes. Finally, cardiac afferents of spinal origin clearly participate in the neural regulation of cardiovascular function. However, it was never our intention to include these in this review. We have, however, cited some key references (Pg 3 Ln16-17).

Specific comments are below:

Page 4, line 45: Please provide references for this statement.

In this revised version, the section on 'Ventricular vagal afferents' has been rewritten with appropriate references provided for chemical activation (pg 5 Ln 64--68) ventricular distension (pg 5, Ln 69-71) and sympathetic stimulation (Pg 5, Ln 72)

Page 7, line 126: Figure 2 should be figure 1.

Thank you for pointing this out. We have added a new figure per the senior editor's suggestion, so this has been corrected

Page 9, para 2: Here one should discuss the paper by Boettcher et al (AJP, 1982 PMID: 7065218) where it was shown that the Bainbridge reflex was markedly blunted in humans and baboons. In the last paragraph on page 9 and beginning of page 10 it might be useful to discuss two papers by Peterson et al. on volume expansion in primates (PMID:6767667 and 111262).

In this revised version, the section on atrial volume reflexes has been rewritten with discussion of the recommended papers (Pg 7-8, Ln 132 – 134, and Pg 9 Ln 160-162).

Page 11, para 1: The point is well taken here but the paper of Hakumaki and Goetz should be cited (PMID: 3927749).

In this revised version, the section on integrated reflex circulatory control during mild hypovolaemia has been rewritten and the paper by Hakumaki and Goetz cited (Pg 14, Ln 305-308)

Page 11: line 240: I would eliminate the word "overwhelming". This is editorial license and in this reviewer's opinion, the evidence is not overwhelming.

We agree with the reviewer and the word been removed.

Page 12: Dynamic exercise. I think this section should be expanded. Muscle afferent feedback in the regulation of sympathetic outflow generally occurs during static exercise or when blood flow to the limb is restricted.

Thank you for this comment. We have expanded on the section, and addressed some key literature relating to central command, muscle afferent feedback, and baroreflex resetting (Pg 17 Ln 382-394). Furthermore, we have added a study by White et al. 2013) that observed a change in pulmonary artery haemodynamics during static exercise and post exercise circulatory arrest (Pg 18, Ln 429-430).

Page 13, line 278: Figure 3 should be figure 2.

Thank you for pointing this out. We have added a new figure per the senior editor's suggestion, so this has been corrected

Page 14, line 327: This may be true but effects on renal resistance and renal sympathetic nerve activity may be an important role of atrial receptor activation, especially chronically.

Thank you for this comment. In this revised version we have highlighted that the renal effects of atrial volume receptors are likely to play an important role, especially in long term control (Pg 20-21, Ln 483 -488)

Referee #2:

I enjoyed reading the paper by Moore et al which is a well written review of the multiplicity of circulatory afferents involved in cardiovascular control. It is scholarly and thorough, and very informative merging historical data with more recent insights.

Thank you for this positive feedback.

I have a few suggestions that might improve the review that I offer to the authors for consideration.

1). The authors set up the idea of "cardiopulmonary" baroreflexes as a straw man that they plan to knock down, and I understand that in the current climate of AltMetrics and social media, coming up with a provocative title might enhance downloads and attention. And if this paper was an editorial or "on my mind" type of piece, then I would be more accepting of that approach. However it is a review paper in the premier physiology journal, and the majority of the paper appropriately presents a thoughtful integration of science rather than a debate. Therefore I suggest that the authors reframe the paper and change the title to avoid sensationalism and focus on the scholarly nature of their presentation.

We are grateful for the candid observations. As suggested, we have reframed the paper and changed the title.

2). The authors put forth the hypothesis that pulmonary artery baroreceptors contribute to sympathetic activation during exercise and hypoxia. However I am not convinced that these receptors are physiologically meaningful, particularly in relation to other well known inputs, such as skeletal muscle afferents and chemoreceptors. PA pressure does not rise during small muscle mass handgrip exercise, yet it leads to marked sympathetic activation, even during circulatory arrest when there is no muscle contraction. Moreover volume infusion which clearly raises PA pressure is associated with sympathetic silence, as shown by Jim Pawelczyk in the authors cited reference from 2001. I also am not clear how the authors are separating feedback from pulmonary baroreceptors from chemoreflexes, at least quantitatively. I would be very cautious about interpreting the association between PAP and MSNA during hypoxia, when both stimuli are changing simultaneously as representing cause and effect. As only one example, there is a near perfect relationship between max VE in L/min and maxVO₂ in L/min - not because VE determines VO₂, but because both vary with body size. I suggest the authors revisit this formulation, and try to put the role of pulmonary baroreceptors in context.

Thank you for your observations concerning our proposals relating to the physiological role of pulmonary artery baroreceptors. The reservations expressed are reasonable, and we understand that we still need to convince the reviewer of the potential of pulmonary artery baroreceptors, relative to other more established inputs. We acknowledge the speculative nature, but it is grounded in our observations in laboratory animals and re/interpretation of experimental data collected in humans.

Exercise. Balanos, White and colleagues (2008, 2013) report that pulmonary arterial systolic pressure (PASP) is elevated, by around 20%, during static lower limb exercise (PMID 18218964, PMID23064507). Furthermore, despite cardiac output returning to rest values, the increase in PASP is sustained during circulatory arrest, possibly because of pulmonary vasoconstriction via the metaboreflex. Although, this may open a “can of worms”, the observation of elevated PA pressure during static exercise and circulatory arrest is at odds with the Reviewer’s suggestion that PA pressure does not rise.

Volume infusion. Findings by Pawelczyk et al (2001) are an important consideration because data presented in that study appear to be at odds with an excitatory input from pulmonary baroreceptors. However, rapid volume infusion is rather indiscriminate in the context of stimulation of cardiovascular reflexogenic areas. Hence, suppression of MSNA during rapid volume probably reflects the *net* response to several inputs, especially powerful aortic arch, and carotid sinus baroreceptors.

Chronic high-altitude hypoxia. The reviewer raises an important question concerning interaction with the peripheral chemoreflex., A handful of studies at high altitude find that blunting peripheral chemoreflex drive (breathing 100% O₂ or infusion of dopamine) does not reduce MSNA to anything resembling that under normoxic (sea-level) conditions. It is based on this evidence, that we separate feedback from pulmonary baroreceptors and from chemoreflexes. This possibility is something that we addressed in another review (Simpson et al., 2021 PMID: 33345334). To make this clearer in the present manuscript, we have added these studies to our review (Pg 20 Ln 466-468) and removed reference to an association between PAP and MSNA during hypoxia. Although the data is limited, we

speculate that the cause of high-altitude excitation varies with time and the role of the peripheral chemoreflex becomes attenuated as part of the acclimatization process.

3). I think the authors should also consider that ventricular mechanoreceptors are not necessarily sensitive only to ventricular pressure. So as the heart twists and translates as it contracts, INTRAMYOCARDIAL pressure can go up quite high, clearly well above systolic arterial pressure. So I am not sure that the "normal physiological range" for these receptors is below the activating pressures for ventricular mechanoreceptors. It is also well known that common stimuli like ischemia, or injection of contrast dye will stimulate ventricular afferents, not just external chemical irritants, so these receptors are often responsible for hypotension during myocardial infarction, or coronary angiography. Please see work by Ron Victor who did a variety of studies in the cath lab examining responses in different vascular beds in innervated and denervated hearts to injection of contrast dye.

This is very important point. We have addressed the possibilities raised in the revised manuscript and cited a study by Dr Victor and colleagues (Scherrer et al, 1990 Pg 15, ln 336).

4). The authors use a lot of space to distinguish between myelinated and unmyelinated fibers. I understand why they do so, but it is never stated explicitly, so might be confusing, especially to a clinically oriented reader.

Thank you for this comment. In this revised manuscript we have tried providing a clearer rationale for how the review is structured, including making a distinction between receptors attached to myelinated and unmyelinated fibres (please see pg 3, ln 18-20).

5). I think the authors neglect some important work supporting the idea that it is pulsatile distortion or arterial baroreceptors by SV that play a critical role in regulation of sympathetic activity during changes in cardiac filling. This is true after spaceflight where both plasma volume and ventricular remodeling occur, and also in patients with continuous flow left ventricular assist devices. The paper by Cornwell et al (Circulation. 2015 Dec 15;132(24):2316-22) is remarkable in that it shows that an intervention that INCREASES mean arterial pressure (and also coronary perfusion pressure) but reduces pulsatility leads to INCREASES in MSNA. I'd like to learn the authors thoughts on how these studies inform their conceptual framework

Thank you for this comment. In the revised manuscript we have highlighted the role that changes in stroke volume have on pulse amplitude and pressure. We have cited studies in astronauts (see pg 13, ln 292-294) and patients with continuous flow LVADs (see pg 14, ln 295-303). We agree that these works support the view that pulsatility changes in the aorta and carotid sinuses, rather than reduced cardiac filling, are the cause of reflex sympathoexcitation observed during mild LBNP.

The study by Dr Cornwell and colleagues is unique because the stimulus to the baroreceptors is isolated, to some degree, from negative feedback. This resembles vascular isolation and perfusion in laboratory animals. Thus, downstream changes in MAP do not feedback to the baroreceptors, increasing the likelihood that MSNA responses are not 'buffered'. With reference to pressure changes in the coronary circulation, although the pressure pattern in the aorta and coronary arterial system were not measured, it seems reasonable to conclude that changes in coronary perfusion pulsatility and receptor distortion also will have ensued. However, coronary baroreceptors have a very low operating range and do not respond to changing the pulsatility in the same way as carotid baroreceptors (McMahon et al, 1996b). Thus, two scenarios appear possible: 1) distortion of

coronary baroreceptors was affected in a fashion like systemic baroreceptors and this contributed to MSNA changes or 2) pump speed changes had little effect on coronary baroreceptors, because the instantaneous value of pressure would remain above the threshold for activation.

6). I think the authors are too dismissive of the idea that sympathetic withdrawal occurs during some patients with syncope. As noted above, even if the heart is not empty, there is often marked increases in contractility during hypovolemia which alters intramyocardial pressure even if the walls of the heart are not smacking against each other. Fu et al (J Physiol. 2012 Apr 15;590(8):1839-48) have published a variety of different pathways that syncope may occur in healthy individuals, including those with and without sympathetic withdrawal. There are multiple pathways by which this kind of neurally mediated syncope can occur.

Thank for raising this important point. In the revised manuscript we have gone into more detail about the multiple pathways and cited the work by Dr Fu and colleagues (Pg 15, Ln 342).

7). a few specific points

- Page 3, line 6: what "unphysiological" stimuli are you talking about here?

We were referring to mass stimulation, such as administration of exogenous irritant chemicals, or quite gross mechanical interventions (obstruction to inflow or outflow) that cause stimulation that is greater than that normally found. However, in the revised version we have taken a different approach to make a similar point (please see pg 7 Ln, 116 – 124)

- Page 9, line 185 - how do atrial receptors regulate "heart size"? Do you mean central blood volume? Atrial or ventricular filling?

We agree that this is a vague statement. In our revised submission we sought to clarify our point by speculating that atrial volume receptors contribute to optimal cardiac filling and diastolic volume - through effects of heart rate and renal function – and have rather minimal effect on systemic vascular resistance (Pg 9, Ln 165-178)

ENDS

Dear Dr Moore,

Re: JP-TR-2022-282305R1 "DIFFERENTIAL CONTRIBUTIONS OF CARDIAC, CORONARY AND PULMONARY ARTERY VAGAL MECHANORECEPTORS TO REFLEX CONTROL OF THE CIRCULATION" by Jonathan P Moore, Lydia L Simpson, and Mark J. Drinkhill

Thank you for submitting your revised Topical Review to The Journal of Physiology. It has been assessed by the original Reviewing Editor and Referees and has been well received. Some final revisions have been requested.

The reports are copied at the end of this email. Please address all of the points and incorporate all requested revisions, or explain in your Response to Referees why a change has not been made.

NEW POLICY: In order to improve the transparency of its peer review process The Journal of Physiology publishes online as supporting information the peer review history of all articles accepted for publication. Readers will have access to decision letters, including all Editors' comments and referee reports, for each version of the manuscript and any author responses to peer review comments. Referees can decide whether or not they wish to be named on the peer review history document.

I hope you will find the comments helpful and have no difficulty in revising your manuscript within 2 weeks.

Your revised manuscript should be submitted online using the links in Author Tasks Link Not Available. This link is to the Corresponding Author's own account, if this will cause any problems when submitting the revised version please contact us.

You should upload:

- A Word file of the complete text (including any Tables);
- An Abstract Figure, (with accompanying Legend in the article file)
- Each figure as a separate, high quality, file;
- A full Response to Referees;
- A copy of the manuscript with the changes highlighted.
- Author profile. A short biography (no more than 100 words for one author or 150 words in total for two authors) and a portrait photograph of the two leading authors on the paper. These should be uploaded, clearly labelled, with the manuscript submission. Any standard image format for the photograph is acceptable, but the resolution should be at least 300 dpi and preferably more.

- A 'Cover Art' file for consideration as the Issue's cover image;
- Appropriate Supporting Information (Video, audio or data set https://jp.msubmit.net/cgi-bin/main.plex?form_type=display_requirements#supp).

To create your 'Response to Referees' copy all the reports, including any comments from the Senior and Reviewing Editors into a Word, or similar, file and respond to each point in colour or CAPITALS. Upload this when you submit your revision.

I look forward to receiving your revised submission.

Yours sincerely,

Ian D. Forsythe
Deputy Editor-in-Chief
The Journal of Physiology
<https://jp.msubmit.net>
<http://jp.physoc.org>
The Physiological Society
Hodgkin Huxley House
30 Farringdon Lane
London, EC1R 3AW
UK
<http://www.physoc.org>
<http://journals.physoc.org>

EDITOR COMMENTS

Reviewing Editor:

The authors have taken all comments on board and have improved the review. Again, I enjoyed reading it. There are, however, some minor points that need to be considered, which are highlighted by reviewer 2.

Main comment: The authors need to add some caveats to the evidence discussed that supports a role for the pulmonary baroreceptors in controlling sympathetic nerve activity at high altitude. Reviewer 2 highlights this. Additionally - the data by Fisher et al. 2018 uses dopamine infusions which does causes some systemic vasodilation even at low dose - which would have opposing effects on MSNA versus 'switching off' the carotid body - this makes the data difficult to interpret and thus the lack of decrease in MSNA during low dopamine infusion at high altitude does not necessarily mean that other afferent inputs (i.e the pulmonary baroreceptors) are contributing to elevated MSNA at altitude.

REFEREE COMMENTS

Referee #1:

I think the authors have done a good job revising this review and have been responsive to the reviewers concerns. I have no further comments.

Referee #2:

The review by Moore et al has been substantially improved, and for the most part, I agree with and like all the changes they have made. There are only a couple of points for which I think the authors are wide of the mark.

1). I am not convinced that their hypoxia experiments support an important sympathetic excitatory role of pulmonary baroreceptors. The two points that I made in my previous review remain unanswered, especially in the body of the manuscript. A) during hypoxia, there is stimulation of CHEMORECEPTORS that clearly increase sympathetic nerve activity. Chemoreceptors upregulate during altitude acclimatization which is a critical component of ventilatory acclimatization and persists even after return to sea level. Just like resetting of arterial baroreceptors during exercise which the authors discuss in some detail, in my opinion, it is this upregulation that explains the persistence of sympathetic activation in the Hansen/Sander studies despite the addition of supplemental oxygen, especially since it persisted after descent to sea level and likely reduction in PAP. Please note that they also gave a saline infusion during these studies to restore plasma volume which decreased (not increased which it should have since PA pressure almost certainly went up) but did not silence SNA. The recent publication by Lundby and Sander showing marked elevation of SNA in high altitude natives who likely do not have pulmonary hypertension, argue for a more prominent role of chemoreflex control than pulmonary baroreceptor contribution to sympathetic activation at altitude.

Second I would argue that it is the restoration of blood oxygen content with acclimatization which abolishes the initial systemic vasodilation caused by hypoxia and induces subsequent hypertension at altitude in the face of this chemoreflex upregulation, not pulmonary baroreceptor feed forward sympathetic activation. I don't understand what you mean by "defending" blood pressure during prolonged hypoxia since the acute hypotension is only very transient. B). The authors have not satisfied me with their answer to my question about the compelling evidence that saline infusion, which raises pulmonary pressure acutely, causes sympathetic silence as observed by Pawelczyk in 2001. Although it is true that stroke volume increases, and despite the absence of a major change in mean arterial pressure, there likely is increased baroreceptor stimulation, the influence of pulmonary hypertension must be quite weak to be so easily and completely overridden. Even if the authors don't agree with this interpretation completely, I do think they need to address this obvious hole in their reasoning directly in the paper.

B). I don't think you have it exactly right in your interpretation of the LVAD experiments. On the one hand you are correct that decreasing pump speed increased pulsatility and sympathetic nerve activity while increasing pump speed decreased pulsatility and decreased SNA; but I would argue that BP was increased because of increased cardiac output from increasing flow through the device, perhaps augmented though by the increase in SNA. Anyway, don't ignore the cardiac output! I would also add the work done by George Hajduczuk and Frank Abboud who along with the Chapleau experiments suggested that there were "rheoreceptors" in the carotid bodies.

2). A few minor issues:

- line 10-13; as the authors describe further down this page, I think the prevailing view of "cardiopulmonary" receptors is their sympathetic activation in response to unloading, rather than, or at least in addition to, the inhibitory response to stimulation.

- line 131; "tachycardia" is misspelled

- line 354-356; in the Fu studies, it is likely that in the subjects who had hypotension despite persistence of elevated SNA, that adrenal release of epinephrine caused beta 2 mediated vasodilation and induced the hypotension. I continue to wonder though whether we really understand the nuances of the PATTERN of sympathetic discharge in the setting of syncope. It is possible that integrating the nerve hides important information about individual fiber discharge that might result in less NE

release. Just a thought....

-- line 407-08; although there is not a clear increase in SNA during low level exercise, are you convinced that there is a DECREASE? If so, please include those references...

REQUIRED ITEMS:

-Author profile(s) must be uploaded via the submission form. Authors should submit a short biography (no more than 100 words for one author or 150 words in total for two authors) and a portrait photograph of the two leading authors on the paper. These should be uploaded, clearly labelled, with the manuscript submission. Any standard image format for the photograph is acceptable, but the resolution should be at least 300 dpi and preferably more. A group photograph of all authors is also acceptable, providing the biography for the whole group does not exceed 150 words.

END OF COMMENTS

1st Confidential Review

11-Mar-2022

AUTHOR RESPONSE

EDITOR COMMENTS

Reviewing Editor:

The authors have taken all comments on board and have improved the review. Again, I enjoyed reading it. There are, however, some minor points that need to be considered, which are highlighted by reviewer 2.

Main comment: The authors need to add some caveats to the evidence discussed that supports a role for the pulmonary baroreceptors in controlling sympathetic nerve activity at high altitude. Reviewer 2 highlights this.

Thank you for the positive comments. We are very pleased to have this opportunity to respond to Editor and Reviewer Comments, and to submit a second revision of our review.

Additionally - the data by Fisher et al. 2018 uses dopamine infusions which does causes some systemic vasodilation even at low dose - which would have opposing effects on MSNA versus 'switching off' the carotid body - this makes the data difficult to interpret and thus the lack of decrease in MSNA during low dopamine infusion at high altitude does not necessarily mean that other afferent inputs (i.e. the pulmonary baroreceptors) are contributing to elevated MSNA at altitude.

The vascular action of dopamine complicates interpretation of findings by Fisher et al. Thus, we are cautious about these findings, although we and others (Hansen & Sander 2003, Simpson et al 2019) observed that high altitude sympathoexcitation is not abolished by hyperoxia, an alternative method to acutely silence peripheral chemoreceptor drive. In the revised manuscript we have surfaced our own findings (with 100% O₂) and placed these in context of findings of Hansen and Sander (2003) and Fisher et al (2018). We have added some more detail to the section of the review that addresses the potential role of pulmonary baroreceptors at high altitude (Ln 462 – 478)

REFEREE COMMENTS

Referee #1:

I think the authors have done a good job revising this review and have been responsive to the reviewer's concerns. I have no further comments.

Thank you.

Referee #2:

The review by Moore et al has been substantially improved, and for the most part, I agree with and like all the changes they have made. There are only a couple of points for which I think the authors are wide of the mark.

Thank you for the generally very positive comments. We are grateful to have another opportunity to respond to the Reviewer's challenge and apologize for not doing this do this well in the previous response.

1). I am not convinced that their hypoxia experiments support an important sympathetic excitatory role of pulmonary baroreceptors. The two points that I made in my previous review remain unanswered, especially in the body of the manuscript. A) during hypoxia, there is stimulation of CHEMORECEPTORS that clearly increase sympathetic nerve activity. Chemoreceptors upregulate during altitude acclimatization which is a critical component of ventilatory acclimatization and persists even after return to sea level. Just like resetting of arterial baroreceptors during exercise which the authors discuss in some detail, in my opinion, it is this upregulation that explains the persistence of sympathetic activation in the Hansen/Sander studies despite the addition of supplemental oxygen, especially since it persisted after descent to sea level and likely reduction in PAP. Please note that they also gave a saline infusion during these studies to restore plasma volume which decreased (not increased which it should have since PA pressure almost certainly went up) but did not silence SNA. The recent publication by Lundby and Sander showing marked elevation of SNA in high altitude natives who likely do not have pulmonary hypertension, argue for a more prominent role of chemoreflex control than pulmonary baroreceptor contribution to sympathetic activation at altitude.

We agree that peripheral chemoreflex upregulation is critical for the time-dependent increase in ventilation at high-altitude (i.e., ventilatory acclimatization) (Nielsen et al., 1988; Dempsey et al., 2014;). However, altered peripheral chemoreflex control of sympathetic outflow cannot be inferred from alterations in respiratory drive, despite the common afferent pathway. Furthermore, evidence has emerged demonstrating some dissociation between peripheral and central chemoreflex modulation of respiratory drive and vascular sympathetic outflow (Keir et al., 2019; Prasad et al., 2020). Additionally, whilst microneurographic investigations of the role of the peripheral chemoreflex in high altitude sympathetic activation is limited to a small number of studies (Hansen and Sander, 2003; Fisher et al., 2018; Simpson et al., 2019; Simpson et al., 2020), none of these found complete reversal of sympathetic neural overactivity by interventions (breathing 100 % O₂ or infusion of low dose dopamine) designed to eliminate / diminish peripheral chemoreceptor activation. We acknowledge that this could mean that experimental approaches to diminish / eliminate the peripheral chemoreflex were unsuccessful, or that some other sympathoexcitatory signal is uncovered that compensates for reduced chemoreflex. However, in experimental animals exposed to prolonged hypoxia direct recordings of carotid chemoreceptor discharge demonstrate that afferent activity is completely abolished after 3-4 minutes of hyperoxia (Åstrand, 1954, PMID 13171114), suggesting that chemoreflex drive is diminished during hyperoxia administration and this is a suitable method to determine the contribution of the peripheral chemoreflex to sympathetic outflow at high altitude. Together, these findings suggest that peripheral chemoreflex activation alone cannot underpin sympathetic activation in chronic hypoxia, even if chemoreceptors upregulate for ventilatory acclimatization.

Also, we agree that high altitude sympathoexcitation persists after return to sea level (Hansen and Sander, 2003; Mitchell et al., 2018). However, incomplete reversal of elevated pulmonary artery pressure following descent to sea-level (Maufrais et al., 2017 PMID 28329216), may provide an alternate / additional mechanism to chemoreceptor upregulation.

We have specifically addressed these points in lines 462 to 471 of the revised manuscript.

The reviewer refers to work by Lundby et al. (2017) and suggests sympathetic neural overactivity in high altitude natives in the absence of concurrent pulmonary hypertension (Lundby et al, 2017) contradicts our reasoning. However, we also studied Andeans (Simpson et al., 2021a) and observed

quantitatively similar values for both MSNA and pulmonary arterial systolic pressure compared with lowlanders studied at the same elevation (Simpson et al., 2020). Furthermore, in another highland native population, Nepali Sherpa, we observed markedly lower basal MSNA compared with acclimatizing lowlanders at the same elevation (5050m) (Simpson et al, 2019). In fact, our data indicates that basal sympathetic neural activity in Sherpa is more like lowlanders at sea level and markedly lower than both lowlanders at high altitude and Andeans (Simpson et al., 2021a). Notably, Sherpa display only modest elevation of pulmonary arterial pressure compared with Andeans and lowlanders (Antezana et al., 1998; Hoit et al., 2005). Furthermore, although we have pointed out that there may be dissociation between respiratory and sympathetic vasomotor limbs of the peripheral chemoreflex in lowland natives, it is notable that hypoxic chemosensitivity differs among Andeans and Himalayans. Compared with Tibetans – close relatives of Sherpa - Andeans demonstrate notable hypoventilation and display blunted peripheral chemoreflex sensitivity (Zhuang et al, 1993, Beall, 1997). Despite this Andeans display a high level of basal MSNA, which argues against chemoreflex activation as the prominent cause of sympathetic outflow, at least in these HA populations.

Second, I would argue that it is the restoration of blood oxygen content with acclimatization which abolishes the initial systemic vasodilation caused by hypoxia and induces subsequent hypertension at altitude in the face of this chemoreflex upregulation, not pulmonary baroreceptor feed forward sympathetic activation. I don't understand what you mean by "defending" blood pressure during prolonged hypoxia since the acute hypotension is only very transient. B). The authors have not satisfied me with their answer to my question about the compelling evidence that saline infusion, which raises pulmonary pressure acutely, causes sympathetic silence as observed by Pawelczyk in 2001. Although it is true that stroke volume increases, and despite the absence of a major change in mean arterial pressure, there likely is increased baroreceptor stimulation, the influence of pulmonary hypertension must be quite weak to be so easily and completely over ridden. Even if the authors don't agree with this interpretation completely, I do think they need to address this obvious hole in their reasoning directly in the paper.

We agree that restoration of CaO_2 probably minimizes vasodilator signaling during HA acclimatization. Nevertheless, some residual vasodilation may remain due to lower oxygenation status and the potential release of ATP from erythrocytes during deoxygenation. Indeed, greater decreases in MAP are observed following non-bursts sequences in Lowlanders during HA acclimatization (Berthelsen et al., 2020 DOI: 10.1152/ajpheart.00364.2020), potentially indicating a greater tonic vasodilator drive compared to SL. However, to clarify, we believe that sympathoexcitation during HA acclimatization is important in preventing hypotension, in other words defending blood pressure, not only because it opposes hypoxic vasodilation, but because it compensates for the effects of reduced total blood volume and reduced α -adrenergic receptor responsiveness. We believe that the absence of such marked increases in MSNA (up to 300%) would likely result in a fall in blood pressure during high altitude acclimatization, and the primary aim is to protect against hypotension, at the expense of increases in BP occurring during more prolonged HA exposures.

The reviewer reiterates inconsistency between our proposal and findings by Pawelczyk et al. (2001), as well by Hansen and Sander (2003). Certainly, the highlighted studies observed a reduction in MSNA - by 15 to 35% - during rapid saline infusion. However, neither of these studies was designed to specifically examine a causal stimulus-response relationship between pulmonary arterial pressure and vasomotor responses. Notably, in experimental animals, when it is possible to independently control distending pressures in pulmonary, aortic, coronary, and carotid sinus baroreceptor regions, we found the threshold for the vascular response was around 25mmHg. In addition, we found the

stimulus-vascular response relationship was saturated at around 35 mmHg, indicating that pulmonary arterial baroreceptors have a comparatively narrow operating range. In our view, a plausible explanation for the reduction in MSNA in the example shown in Fig 1 in Pawelczyk et al. (2001) is that the threshold for pulmonary arterial mechanoreceptor mediated sympathetic activation had not been surpassed and; or the pulmonary baroreflex is functioning on the flat portion of the stimulus-response relationship, whereas the systemic arterial baroreceptor reflex mechanism is operating close to maximal sensitivity. Moreover, in the study conducted at high altitude by Hansen and Sander it is probable that the pulmonary baroreceptor stimulus-response relationship was functioning close to the saturation point. Thus, we speculate that changes in pulmonary arterial pressure, as result of rapid infusion, had minimal effect on the PAP-MSNA relationship, whereas increased stroke volume and consequent widening of arterial pulse amplitude could explain the 15 to 35% reduction in MSNA in both studies.

We have highlighted the work by Pawelczyk et al and presented some possible explanations for the divergent findings (Ln 482- 492)

B). I don't think you have it exactly right in your interpretation of the LVAD experiments. On the one hand you are correct that decreasing pump speed increased pulsatility and sympathetic nerve activity while increasing pump speed decreased pulsatility and decreased SNA; but I would argue that BP was increased because of increased cardiac output from increasing flow through the device, perhaps augmented though by the increase in SNA. Anyway, don't ignore the cardiac output! I would also add the work done by George Hajduczuk and Frank Abboud who along with the Chapleau experiments suggested that there were "reoreceptors" in the carotid bodies.

Thank you. We have addressed the effect of increased flow in Ln 305-306 and added the reference by Hajduczuk et al. (1988) in Ln 316.

2). A few minor issues:

- line 10-13; as the authors describe further down this page, I think the prevailing view of "cardiopulmonary" receptors is their sympathetic activation in response to unloading, rather than, or at least in addition to, the inhibitory response to stimulation.

We agree that substantial attention has focused upon neural control from cardiopulmonary receptors during reductions in central volume and pressure. Indeed, challenging this notion is fundamental to this review. Nevertheless, there are many examples in the literature which point to cardiopulmonary receptors as a negative feedback mechanism, analogous to systemic arterial baroreceptors (Bishop, Malliani & Thoren 1983, DOI: 10.1002/cphy.cp020315 Persson 1991, DOI: 10.1007/978-3-642-76366-3_5, Katayama & Saito 2019), and that loading of cardiopulmonary receptors inhibits sympathetic vasomotor discharge in humans (Mancia & Mark, 1983 DOI 10.1002/cphy.cp020320; Raven Young & Fadel, 2019 PMID 30921029).

- line 131; "tachycardia" is misspelled

Misspelling has been corrected. Thank you for pointing this point.

- line 354-356; in the Fu studies, it is likely that in the subjects who had hypotension despite

persistence of elevated SNA, that adrenal release of epinephrine caused beta 2 mediated vasodilation and induced the hypotension. I continue to wonder though whether we really understand the nuances of the PATTERN of sympathetic discharge in the setting of syncope. It is possible that integrating the nerve hides important information about individual fiber discharge that might result in less NE release. Just a thought....

Thank you. We agree that other methods of analysis probably provide more insight into the role of sympathetic control in syncope, and probably other stressors. Thus, we have added a couple to sentences (Ln 369-374) with reference to address this.

-- line 407-08; although there is not a clear increase in SNA during low level exercise, are you convinced that there is a DECREASE? If so, please include those references...

We have highlighted the references (Ln 426).

Dear Dr Moore,

Re: JP-TR-2022-282305R2 "DIFFERENTIAL CONTRIBUTIONS OF CARDIAC, CORONARY AND PULMONARY ARTERY VAGAL MECHANORECEPTORS TO REFLEX CONTROL OF THE CIRCULATION" by Jonathan P Moore, Lydia L Simpson, and Mark J. Drinkhill

I am pleased to tell you that your Topical Review article has been accepted for publication in The Journal of Physiology, subject to any modifications to the text that may be required by the Journal Office to conform to House rules.

NEW POLICY: In order to improve the transparency of its peer review process The Journal of Physiology publishes online as supporting information the peer review history of all articles accepted for publication. Readers will have access to decision letters, including all Editors' comments and referee reports, for each version of the manuscript and any author responses to peer review comments. Referees can decide whether or not they wish to be named on the peer review history document.

The last Word version of the paper submitted will be used by the Production Editors to prepare your proof. When this is ready you will receive an email containing a link to Wiley's Online Proofing System. The proof should be checked and corrected as quickly as possible.

All queries at proof stage should be sent to tjp@wiley.com

The accepted version of the manuscript will be published online, prior to copy editing in the Accepted Articles section.

Are you on Twitter? Once your paper is online, why not share your achievement with your followers. Please tag The Journal (@jphysiol) in any tweets and we will share your accepted paper with our 22,000+ followers!

Yours sincerely,

Ian D. Forsythe
Deputy Editor-in-Chief
The Journal of Physiology
<https://jp.msubmit.net>
<http://jp.physoc.org>
The Physiological Society
Hodgkin Huxley House
30 Farringdon Lane
London, EC1R 3AW
UK
<http://www.physoc.org>
<http://journals.physoc.org>

*** IMPORTANT NOTICE ABOUT OPEN ACCESS ***

To assist authors whose funding agencies mandate public access to published research findings sooner than 12 months after publication The Journal of Physiology allows authors to pay an open access (OA) fee to have their papers made freely available immediately on publication.

You will receive an email from Wiley with details on how to register or log-in to Wiley Authors Services where you will be able to place an OnlineOpen order.

You can check if your funder or institution has a Wiley Open Access Account here <https://authorservices.wiley.com/author-resources/Journal-Authors/licensing-and-open-access/open-access/author-compliance-tool.html>

Your article will be made Open Access upon publication, or as soon as payment is received.

If you wish to put your paper on an OA website such as PMC or UKPMC or your institutional repository within 12 months of publication you must pay the open access fee, which covers the cost of publication.

OnlineOpen articles are deposited in PubMed Central (PMC) and PMC mirror sites. Authors of OnlineOpen articles are permitted to post the final, published PDF of their article on a website, institutional repository, or other free public server, immediately on publication.

Note to NIH-funded authors: The Journal of Physiology is published on PMC 12 months after publication, NIH-funded authors DO NOT NEED to pay to publish and DO NOT NEED to post their accepted papers on PMC.

EDITOR COMMENTS

Reviewing Editor:

No further comments.

Senior Editor:

Thank you for these final revisions. I look forward to seeing your article in press.

REFEREE COMMENTS

Referee #2:

The authors have done a good job responding to the previous comments. Although there remain some areas of disagreement, I don't think they will be resolved by continued revision. For the authors, I would be cautious in interpreting the various data coming out of the Saito lab, some of which was acquired in my lab in the early 90s. For example, in his 1997 paper cited by the authors, it states:

"To investigate the effects of exercise duration on muscle sympathetic nerve activity (MSNA), heart rate, blood pressure (BP), tympanic temperature, blood lactate concentration, and thigh electromyogram were measured in eight volunteers during 30 min of cycling in the sitting position at an intensity of 40% of maximal oxygen uptake. MSNA burst frequency increased 18 min after exercise was begun (25 +/- 4 bursts/min at baseline and 36 +/- 5 bursts/min at 21 min of exercise), reaching 41 +/- 5 bursts/min at the end of exercise."

Although there may be some variation in MSNA at the very lowest levels of external work (say 20% of VO₂max), I don't think it would be universal to note that MSNA "decreases" during exercise.

In any case, nice job and this will be an important contribution.

2nd Confidential Review

27-Jun-2022